# A Techno-Economic Assessment of Fischer–Tropsch Fuels Based on Syngas from Co-Electrolysis

Ralf Peters [1,*] , Nils Wegener [1,2], Remzi Can Samsun [1] , Felix Schorn [1,3] , Julia Riese [2] , Marcus Grünewald [2] and Detlef Stolten [3,4]

1 Institute of Energy and Climate Research-Electrochemical Process Engineering (IEK-14), Forschungszentrum Jülich GmbH, Wilhelm-Johnen-Str., 52428 Jülich, Germany; n.wegener@fz-juelich.de (N.W.); r.c.samsun@fz-juelich.de (R.C.S.); f.schorn@fz-juelich.de (F.S.)

2 Laboratory for Fluid Separations, Faculty of Mechanical Engineering, Ruhr University Bochum, 44721 Bochum, Germany; julia.riese@fluidvt.ruhr-uni-bochum.de (J.R.); marcus.gruenewald@fluidvt.ruhr-uni-bochum.de (M.G.)

3 Institute of Energy and Climate Research-Techno-Economic System Analysis (IEK-3), Forschungszentrum Jülich GmbH, Wilhelm-Johnen-Str., 52428 Jülich, Germany; d.stolten@fz-juelich.de

4 Chair for Fuel Cells, RWTH Aachen University, 52072 Aachen, Germany

* Correspondence: ra.peters@fz-juelich.de; Tel.: +49-2461-61-4260

**Abstract:** As a part of the worldwide efforts to substantially reduce $CO_2$ emissions, power-to-fuel technologies offer a promising path to make the transport sector $CO_2$-free, complementing the electrification of vehicles. This study focused on the coupling of Fischer–Tropsch synthesis for the production of synthetic diesel and kerosene with a high-temperature electrolysis unit. For this purpose, a process model was set up consisting of several modules including a high-temperature co-electrolyzer and a steam electrolyzer, both of which were based on solid oxide electrolysis cell technology, Fischer–Tropsch synthesis, a hydrocracker, and a carrier steam distillation. The integration of the fuel synthesis reduced the electrical energy demand of the co-electrolysis process by more than 20%. The results from the process simulations indicated a power-to-fuel efficiency that varied between 46% and 67%, with a decisive share of the energy consumption of the co-electrolysis process within the energy balance. Moreover, the utilization of excess heat can substantially to completely cover the energy demand for $CO_2$ separation. The economic analysis suggests production costs of 1.85 €/$l_{DE}$ for the base case and the potential to cut the costs to 0.94 €/$l_{DE}$ in the best case scenario. These results underline the huge potential of the developed power-to-fuel technology.

**Keywords:** $CO_2$ electrolysis; co-electrolysis; electrofuels; power-to-fuel; power-to-liquid; solid oxide electrolysis; synthetic diesel; synthetic kerosene; water electrolysis

## 1. Introduction

In order to mitigate anthropogenic climate change, a considerable reduction in the emission of climate-damaging emissions is necessary. For this purpose, it is essential to either electrify sectors or convert them to the use of alternative fuels in order to minimize dependence on fossil fuels. As a result of the introduction of various sustainable technologies, the energy and household sectors, for example, have seen the first reductions in greenhouse gas emissions [1]. The reduction of emissions in the transport sector, however, presents a bigger challenge. Heavy haulage, ship, and air traffic, in particular, can only be converted to electrified drivetrains to a limited extent. As a result, it can be expected that the demand for liquid fuels with high energy densities will remain high in the future [2]. One way of achieving $CO_2$-neutrality for the transport sector involves the power-to-liquid concept. 'Power-to-liquid' is a collective term for various technologies employed in the production of liquid energy carriers through the use of renewable electrical energy with the addition of carbon dioxide ($CO_2$) [3]. The energy carriers produced in this way can be

converted back into electricity and thus function as electricity storage systems that can be used for other applications. In the event that the energy sources are used as fuels in the transport sector, this is referred to as 'power-to-fuel'. If both the required electrical energy and $CO_2$ are obtained from renewable sources, fuels can be produced in a $CO_2$-neutral manner using power-to-fuel processes. Thus, these represent a possibility for effectively defossilizing the transport sector [4]. Various power-to-fuel concepts already exist. On one hand, the production of alternative fuels such as methanol or dimethyl ether (DME) is a focus of research [5]. However, on the other hand, researchers are also exploring the synthesis of traditional fuels such as gasoline, kerosene, and diesel, as the existing infrastructure and greater applicability of these represent an advantage over alternatives [6]. For the structured processing of these tasks, this paper was divided into the following sections:

- Section 2 provides some insights into the motivation to apply power-to-fuel processes. An overview of already implemented and future planned power-to-liquid or power-to-fuel projects is given.
- Section 3 explains the basics of the individual components used in the developed power-to-fuel process. At the end, the degree of technological maturity of the individual components of the developed fuel synthesis is examined.
- The topic of Section 4 is the development and design of the selected power-to-fuel process. For this purpose, the more precise framework conditions and resulting structure of the fuel synthesis are presented first. Then, the modeling and simulation of the process in Aspen Plus is explored.
- Section 5 presents and discusses the results of the process analysis simulations. First, it is determined whether the fuels produced meet all of the requirements and then the material and energy balance of the process is examined. Finally, based on the efficiency, an energetic comparison between the developed fuel synthesis and alternative power-to-liquid processes is carried out.
- Section 6 analyzes the economic aspects of the developed power-to-fuel process. First, the manufacturing costs of the fuels produced are determined. Then, the influence of various factors on the production costs is examined through a sensitivity analysis. Finally, the production costs of the developed fuel synthesis route are compared with those of alternative power-to-liquid processes.
- In Section 7, the results of the work are summarized and an outlook on the main research areas are given.

## 2. Background

In 2012, the German transport sector consumed 2772 PJ ($\approx$770 TWh) of energy. Around 26.9% of this energy requirement was accounted for by gasoline, 51% by diesel, and 15.7% by aviation fuels [7].

On one hand, as vehicles with alternative drivetrains such as battery- or fuel cell-based ones are increasingly being used, it is expected that gasoline will lose its importance in the long term. On the other hand, due to the lack of alternatives in freight and air traffic, it can be assumed that the fuels diesel and kerosene will also be of great importance over the longer term [2].

Accordingly, the production of renewable diesel and kerosene is of both academic and industrial interest. A major advantage of synthetically-produced diesel and kerosene is that they are compatible with existing infrastructures and can be used in current vehicles [4]. The prerequisite for use is the fulfillment of the fuel specifications, which are set out in the relevant standards. Synthetically-produced diesel must comply with EN 15940 in Europe. ASTM 7566 applies to Jet A-1 kerosene used in civil aviation. This allows conventionally-produced Jet A-1 to be mixed with up to 50% synthetically-produced kerosene, depending on the synthesis route [8]. This synthetic kerosene is called synthesized paraffinic kerosene (SPK). An extract of the most important parameters for diesel (class A) according to EN 15940 and for SPK produced via a Fischer–Tropsch synthesis according to ASTM 7566 is discussed in more detail in Section 5.1. Class A describes diesel with an increased cetane

number, which is a characteristic value of the ignitability of diesel fuels. The higher the cetane number, the more readily ignitable the diesel fuel [9].

One possibility of producing renewable, synthetic fuels is via the power-to-fuel concept. Schemme et al. [10] discussed the power-to-fuel concept as a solution to the present challenges of the transport sector in terms of the energy transition by coupling the energy and transport sectors. According to this concept, renewable electricity is used to produce hydrogen via water electrolysis, offering a storage possibility for volatile renewable energy sources. In the following synthesis step, renewable fuels are synthesized in a reaction or a series of reactions and further treatment steps combining the produced hydrogen with carbon dioxide from various possible sources. Different electrolysis technologies can be utilized for the generation of hydrogen.

In 2014, Sunfire GmbH commissioned the "Fuel 1" demonstration plant in Dresden, Germany. At this facility, high-temperature water electrolysis (SOEC) is used to provide hydrogen. The hydrogen is then mixed with carbon monoxide, which is generated in a reverse water–gas shift reactor and converted into so-called blue crude by means of Fischer–Tropsch synthesis. Blue crude is a renewable crude oil that can be further processed in a conventional refinery into synthetic gasoline, kerosene, or diesel, for example [11]. The plant was run for 1500 h [12] and produced one barrel (159 L) of blue crude per day [13].

In 2017, the VTT Technical Research Center of Finland and the Lappeenranta University of Technology operated a demonstration plant in Lappeenranta (Finland) for around 300 operating hours as part of the "SOLETAIR" project [14]. Fischer–Tropsch synthesis was also used in this system. $CO_2$ was obtained through direct air capture and converted into carbon monoxide in a reverse water–gas shift reactor. The hydrogen was provided via PEM electrolysis [15].

As part of the Kopernikus project "Power-to-X", funded by the Federal Ministry of Education and Research, the so-called SUNFIRE-SYNLINK was put into operation in Karlsruhe (Germany) in 2019 [16]. A co-electrolysis system from Sunfire GmbH was used to produce synthesis gas. This was combined with a Fischer–Tropsch reactor from INERATEC and a hydrocracker unit from the Karlsruhe Institute of Technology to produce synthetic fuels. The $CO_2$ required was obtained using Climeworks' direct air capture (DAC) technology. The co-electrolysis currently in use has an output of 10 kW, but Sunfire plans to upscale the process to an industrial scale [17].

An industrial-scale plant is being planned by the Norwegian company, Nordic Electrofuel (formerly Nordic Blue Crude). The plant is to be built and commissioned in Herøya Industripark (Norway) by 2022. The use of high-pressure alkaline electrolysis is planned and the plant is set to achieve an initial production capacity of 10 million liters per year. The required $CO_2$ is to be supplied from both industrial sources and via DAC technology. Originally, the use of a co-electrolysis unit from Sunfire GmbH was planned for the plant planned by Nordic Electrofuel [17]. However, the business partners separated in 2020 and Sunfire GmbH founded the industrial consortium, Norsk e-Fuel [18,19], together with Climeworks, Paul Wurth (SMS Group), and Valinor. Norsk e-Fuel also plans to build a plant capable of producing 10 million liters of synthetic kerosene per year at Herøya Industrial Park (Norway). This facility is scheduled to be commissioned in 2023 and expanded to a production capacity of 100 million liters of renewable fuels by 2026 [20]. Amongst other things, the co-electrolysis technology from Sunfire GmbH and the DAC technology from Climeworks are to be used in it [18,20].

In Rotterdam, the Hague Airport announced a study in 2019 in which, in collaboration with several European partners, a demonstration plant for the production of aviation fuel was to be developed. The plant is expected to produce around 1000 L of renewable fuel, but there has not yet been a specific date for its commissioning [21]. Based on this study, the two startups Synkero and Zenid were presented on 8 February 2021 [22,23]. Both of these plan to build a plant for the production of synthetic kerosene, but each is pursuing different concepts for providing the required $CO_2$. Although Zenid's goal is a plant that obtains

the $CO_2$ exclusively by means of DAC technology [24], Synkero also considers other $CO_2$ sources such as industrial exhaust gases or biogenic sources [25].

Figure 1 illustrates a special power-to-fuel concept for the production of synthetic fuels, which is to be examined in more detail within the scope of this work. Initially, synthesis gas consisting of hydrogen ($H_2$) and carbon monoxide (CO) is produced from water and $CO_2$ using renewable electrical energy. The resulting synthesis gas is then converted into liquid fuels.

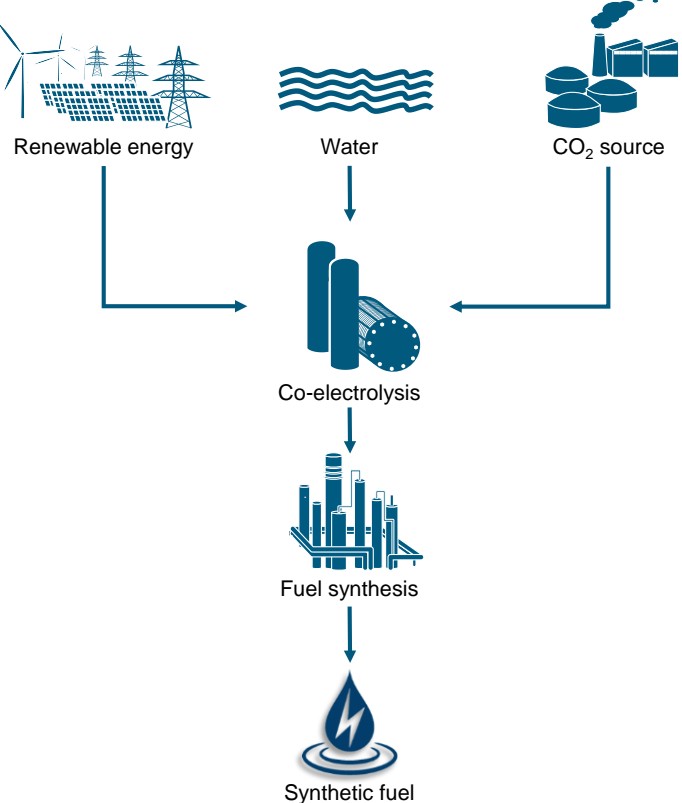

**Figure 1.** Schematic of a power-to-fuel process with co-electrolysis.

What is special about the power-to-fuel concept shown in Figure 1 is that the synthesis feed gas is produced using what is known as co-electrolysis. This allows for the production of synthesis gas in a single step, and so there is no need to produce hydrogen and carbon monoxide separately. In addition, co-electrolysis offers some energetic advantages over other synthesis gas production routes. For example, it is possible to substitute some of the electrical energy required for electrolysis with excess thermal energy from fuel synthesis. Therefore, the combination of co-electrolysis with Fischer–Tropsch synthesis represents an interesting basis for power-to-fuel processes. In this route, synthesis gas is converted into hydrocarbons by means of strongly exothermic reactions, which are then processed into fuels such as gasoline, diesel, or kerosene. The heat of reactions that occur during Fischer–Tropsch synthesis can be used to operate co-electrolysis, which opens up the possibility of improving the overall efficiency of power-to-fuel processes.

The aim of this work was to develop a power-to-fuel process based on co-electrolysis, in combination with a Fischer–Tropsch synthesis, and to model and simulate the process in the Aspen Plus program. The entire process chain, starting with $CO_2$ and water through to fuel, should be considered. Then, the developed power-to-fuel process should be techno-economically analyzed and compared to alternative power-to-fuel processes.

## 3. Basic Process Units of PtF System Design

The following section presents the technical basics of the most important components of the modeled power-to-fuel process. The power-to-fuel system developed within the

scope of this work consists of a water electrolysis, co-electrolysis, Fischer–Tropsch synthesis, hydrocracker, reformer, and carrier steam distillation. This section deals with the basics of these system components.

### 3.1. Electrolysis and Co-Electrolysis

Depending on the electrolyte or ionic charge carrier used, a distinction is made between three different electrolysis methods. Alkaline electrolysis and PEM electrolysis are already available on the megawatt scale, and hydrogen thus produced can achieve high purities of over 99% [6,26]. In addition, alkaline and PEM electrolysis can be operated under pressures of 60 to 80 bar, thus reducing the need for compressor power for downstream processes [6].

In contrast, the solid oxide electrolyzer cell (SOEC) is not yet commercially available in the megawatt range. Operation under pressure is also still in the development phase. However, due to their high operating temperature, SOECs offer thermodynamic advantages over other electrolysis types. Therefore, it is theoretically possible to achieve electrical efficiencies of over 100% based on the calorific value of the products [6]. Typical operating temperatures of a SOEC are 800–1000 °C [27] or 700–1000 °C [28], depending on the literature source. In addition, due to its high operating temperature, it has improved kinetics [29] and can be operated as a so-called co-electrolysis unit [30]. In co-electrolysis, not only water, but also $CO_2$ is broken down. This electrolysis process is especially interesting for power-to-liquid (PtL) and power-to-fuel (PtF) processes as it makes it possible to produce synthesis gas consisting of hydrogen and carbon monoxide in a single process step [6,30].

The electrochemical reactions that take place at the cathode and anode are as follows. Cathode:

$$H_2O + 2e^- \rightarrow H_2 + O^{2-} \tag{1}$$

$$CO_2 + 2e^- \rightarrow CO + O^{2-} \tag{2}$$

Anode:

$$2O^{2-} \rightarrow O_2 + 4e^- \tag{3}$$

The minimum energy expenditure for the decomposition of water or $CO_2$ corresponds to the enthalpy of reaction $\Delta H_R$, which in the case of the usual operating conditions of co-electrolysis (860 °C, 1 bar) is 249 kJ/mol for water and 283 kJ/mol for $CO_2$ [30]. According to the second law of thermodynamics, the reaction enthalpy is composed of the free Gibbs energy $\Delta G_R$ and the reaction entropy $\Delta S_R$, multiplied by the temperature $T$ as follows [31]:

$$\Delta H_R = \Delta G_R + T \cdot \Delta S_R \tag{4}$$

Here, $\Delta G_R$ is the part of the reaction enthalpy that must be provided in the form of electricity during electrolysis, whereas $T \cdot \Delta S_R$ can be supplied to the reaction in the form of heat [32]. An advantage of high-temperature electrolysis compared to other electrolysis types is that if the water evaporates outside the electrolysis cell, this energy must no longer be introduced into the cell in the form of electricity [29]. After evaporation with increasing temperature, the total energy requirement of the reaction remains almost constant, but the amount of electrical energy that is absolutely necessary significantly decreases. On the basis of these two facts, valuable electrical energy can be saved with high-temperature electrolysis in comparison to, for example, PEM electrolysis. Considering the thermal energy that is exchanged, the internal thermal losses are of particular importance in electrolysis, as they can be used to provide the heat of the reaction. The case in which the thermal losses correspond precisely to the heat of the reaction is referred to as the thermoneutral operating point, and the electrical voltage applied to the SOEC at this operating point is correspondingly referred to as the thermoneutral voltage [3]. At the thermoneutral operating point, the entire electrical energy $E_{el}$ supplied to the electrolysis cell is converted

into chemical potential energy and the following applies to cell efficiency at thermoneutral point $\eta_{cell,TN}$ [31]:

$$\eta_{cell,TN} = \frac{\Delta H_R}{E_{el}} = 1 \tag{5}$$

Due to the simultaneous presence of $H_2O$, $CO_2$, $H_2$, and CO at the cathode of the SOEC and the high operating temperatures, in addition to the reactions listed in Equations (1)–(3), the so-called water–gas shift reaction (WGS) or reverse water–gas shift reaction (RWGS) must also be considered in the co-electrolysis [33]. The reverse water–gas shift reaction is favored at the high operating temperatures of a SOEC [34].

$$CO + H_2O \leftrightarrow CO_2 + H_2 \tag{6}$$

In addition, methanation reactions (Equations (7) and (8)) can occur at the cathode of a SOEC [33]:

$$CO + 3H_2 \leftrightarrow CH_4 + H_2O \tag{7}$$

$$CO_2 + 4H_2 \leftrightarrow CH_4 + 2H_2O \tag{8}$$

Finally, the Boudouard reaction (Equation (9)) must be taken into account, and can lead to the precipitation of solid carbon under certain operating conditions of a SOEC [33]:

$$2CO \leftrightarrow CO_2 + C_{(s)} \tag{9}$$

According to Equation (5) the efficiency of co-electrolysis at the thermo-neutral operating point corresponds to 100%. According to Peters et al. [22], however, SOECs are not usually operated with an exactly thermoneutral voltage, so the heat must be either added or removed. In addition, the efficiency of the entire system is influenced by other factors such as the power required for the compression and storage of the products or the losses of the voltage converter to rectify the alternating current. Therefore, in order to assess the efficiency of the overall system ($\eta_{SOEC}$), the degree of efficiency is usually defined through the calorific value of the product (*Mass flow rate of product $m_{product}$ multiplied by its lower heating value $H_u^0$*) in relation to the electrical ($E_{el,\text{total}}$) and thermal energy ($E_{th,\text{total}}$) used [22]:

$$\eta_{SOEC} = \frac{m_{product} \cdot H_u^0}{E_{el,\text{total}} + E_{th,\text{total}}} \tag{10}$$

### 3.2. Fischer–Tropsch Synthesis

Fischer–Tropsch synthesis is a process in which a synthesis gas consisting of hydrogen and carbon monoxide is converted into liquid hydrocarbons [9]. This results in a wide range of hydrocarbon chains of different lengths with chain length n, according to the following equation [33]:

$$(2n + 1) \cdot H_2 + n \cdot CO \rightarrow C_nH_{2n+2} + n \cdot H_2O \tag{11}$$

Another reaction that occurs in the Fischer–Tropsch synthesis process is the water–gas shift reaction already described in Equation (6). The methanation reactions (Equations (7) and (8)) and the Boudouard reaction described in Equation (9) can also occur [35]. Which reactions take place to which extent during the Fischer–Tropsch synthesis and also the product distribution of it are determined by the process parameters [35].

A distinction is made between high-temperature Fischer–Tropsch synthesis (HTFT) and low-temperature Fischer–Tropsch synthesis (LTFT) [36]. High temperatures (300–350 °C), with iron as a catalyst, favor short chain lengths and therefore shift the product distribution in the direction of (liquid) gases ($n = 1$–4) and synthetic gasoline ($n = 5$–12) [6,35]. Lower temperatures (200–240 °C) in combination with iron or cobalt catalysts, in contrast, favor the formation of longer hydrocarbon chains (i.e., middle distillates such as kerosene ($n = 8$–16) [37] and diesel ($n = 10$–23) as well as long-chain ones ($n > 22$)) [6,35]. In addition,

cobalt catalysts suppress the water–gas shift reaction, and low temperatures reduce the formation of methane and solid carbon. The operating pressure also has a direct influence on the product distribution [35]. Typical operating pressures in Fischer–Tropsch synthesis are between 1 and 40 bar, with higher pressures resulting in longer average hydrocarbon chain lengths [35]. Another influencing factor on Fischer–Tropsch synthesis is the ratio of $H_2$ to CO. Typically, $H_2$/CO ratios of around two are used, with the average chain length of the product decreasing with higher ratios and increasing with lower ones [33].

The product distribution can be approximately determined using the Anderson–Schulz–Flory distribution [36]. With this, both the mass fractions $w_n$ (Equation (12)) and molar fractions $x_n$ (Equation (13)) of the respective hydrocarbon chains with chain length n can be determined [35]:

$$w_n = \alpha^{n-1} \cdot (1-\alpha)^2 \cdot n \tag{12}$$

$$x_n = \alpha^{n-1} \cdot (1-\alpha) \tag{13}$$

Here, $\alpha$ stands for the chain growth probability, which is determined by reactor design and operating conditions. The chain growth probability can be either determined empirically or taken from the literature. The influence of chain growth probability on the product distribution is discussed further in Appendix A. In Vervloet et al. [38], the approach in Equation (14) was given for $\alpha$ for the low-temperature Fischer–Tropsch synthesis assumed in this work and the use of a cobalt catalyst. In this model, the chain growth probability is determined through the ratio between the chain growth rate and chain growth termination rate:

$$\alpha = \frac{1}{1 + k_\alpha \left(\frac{c_{H2}}{c_{CO}}\right)^\beta \exp\left(\frac{\Delta E_\alpha}{R}\left(\frac{1}{493.15} - \frac{1}{T}\right)\right)} \tag{14}$$

where

$k_\alpha$ is the ratio of the speeds of the chain growth rate and chain growth termination rate ($k_\alpha = 0.0567$);
$c_{H2}$ is the hydrogen concentration in mol/m$^3$;
$c_{CO}$ is the carbon monoxide concentration in mol/m$^3$;
$\beta$ is the exponential parameter for selectivity ($\beta = 1.76$);
$\Delta E_\alpha$ is the difference of activation energies for chain growth and chain growth termination ($\Delta E_\alpha = 120.4 \frac{kJ}{mol}$);
$R$ is the ideal gas constant ($R = 8.314 \frac{J}{mol}$); and
$T$ is the reactor temperature in K.

The influence of the chain growth probability on the product distribution is illustrated in (Figure A1 in Appendix A, which shows the product distribution of a Fischer–Tropsch synthesis for $\alpha = 0.88$ and $\alpha = 0.92$ for $C_1$ to $C_{60}$.

### 3.3. Hydrocrackers

Hydrocracking refers to a chemical process in which long-chain, higher-molecular hydrocarbon chains are split into shorter ones through the addition of hydrogen. The distribution of the chain lengths of the products is strongly influenced by the catalyst used and the selected process conditions. Therefore, these must always be adapted to the respective application [39,40]. For power-to-fuel processes, chain lengths in the range from $n = 5$ to $n = 20$ are of great importance, as these hydrocarbon chains are required for the production of synthetic gasoline, kerosene, and diesel [36]. One possibility for maximizing these fractions in the product of the hydrocracker is the use of so-called "ideal hydrocracking" [41,42]. As a power-to-fuel process is to be modeled and simulated within the scope of this work, the focus in the following was on ideal hydrocracking.

The most important properties of ideal hydrocracking are defined as follows, drawing on Bouchy et al. [41]. If $C_n$-molecules are cracked, the selectivity to all $C_4$- to $C_{n-4}$ hydrocarbons is identical, the selectivity to $C_3$ and $C_{n-3}$ is half of that, and $C_1$, $C_2$, $C_{n-1}$

and $C_{n-2}$ cannot be formed. Furthermore, only primary cracking occurs. Exclusively primary hydrocracking means that the shorter hydrocarbons that are created after a longer hydrocarbon chain has been cracked cannot be cracked any further [39]. In the case of non-ideal hydrocracking, the proportion of middle distillates is significantly lower than in ideal hydrocracking. In addition, a large peak of the $C_3$ to $C_5$ hydrocarbons was identified by Wegener [43]. This course is due to the occurrence of secondary cracking, which cracks the middle distillates, with the proportion of short-chain hydrocarbons increasing.

According to Bouchy et al., ideal bifunctional catalysts with a hydrogenation/dehydrogenation function and an acid function can be used, whereby it must be ensured that the reaction taking place at the acid function is the limiting one [41]. In addition, it is important to ensure that the pore structure of the catalyst is correct, so that no undesired increased cracking occurs at the ends of the hydrocarbon chains, which can lead to the stronger formation of short-chain gases [41,44].

As already described, the operating conditions also have a major influence on the product distribution of a hydrocracker. Conventional hydrocracking, depending on the literature source, is carried out at temperatures ranging from 350–430 °C and pressures of 100–200 bar [41], or in the upper range of 290–445 °C and 10–200 bar [40]. Ideal hydrocracking takes place under significantly milder process conditions, with temperatures in the range of 324–372 °C and pressures of 35–70 bar [41].

In addition, in order to suppress soot formation and catalyst deactivation, it must be ensured that the proportion of $H_2$ in the feed stream is high enough [39]. In the literature, values of 6–15% by weight are recommended [45]. With hydrocracking, conversions of up to 99% can be theoretically achieved [39]. However, according to Bouchy et al. [41], conversions that are too high for ideal hydrocracking become problematic, as secondary cracking inevitably occurs, even with ideal hydrocracking at very high conversions.

### 3.4. Reformers–Steam Reforming and Partial Oxidation

Reformers make it possible to convert hydrocarbons into synthesis gas. Various reactions and side reactions take place simultaneously in a reformer, with steam reforming and partial oxidation playing the greatest role [4].

In steam reforming, hydrocarbons are converted into carbon monoxide and hydrogen with the addition of steam. The general reaction equation for steam reforming is as follows:

$$C_nH_{2n+2} + n \cdot H_2O \rightarrow n \cdot CO + (2n+1) \cdot H_2 \tag{15}$$

This is a strongly endothermic reaction which, with the exception of methane ($n = 1$), can be regarded as irreversible at the normal operating temperatures of over 500 °C for reformers [46]. The partial oxidation of hydrocarbons is an exothermic reaction with the general reaction equation:

$$C_nH_{2n+2} + \frac{n}{2} \cdot O_2 \rightarrow n \cdot CO + (n+1) \cdot H_2 \tag{16}$$

If the supply of oxygen is regulated, the degree of reaction of the partial oxidation can also be adjusted. Accordingly, if both reactions are carried out at the same time, the required heat of reaction for the steam reforming can be provided via partial oxidation. A reformer can be operated endothermically, exothermically, or autothermically through the oxygen supply in the overall balance [46]. As already noted, numerous side reactions occur in a reformer. Due to the high operating temperatures and the simultaneous presence of water, hydrogen, carbon monoxide, and $CO_2$, the water–gas shift reaction (Equation (6)) takes place [46]. In addition, soot can form due to various reaction mechanisms. These reactions are, in particular, the Boudouard reaction (Equation (9)), methane splitting (Equation (17)), and CO or $CO_2$ hydrogenation (Equations (18) and (19)) [46]. These soot formation reactions are undesirable during operation of the reformer, and can be suppressed by means of a

suitable starting material composition. According to Rostrup-Nielsen [47], $H_2O/C$ rates of 0.6 are suitable for this:

$$CH_4 \leftrightarrow H_2 + C_{(s)} \tag{17}$$

$$CO + H_2 \leftrightarrow H_2O + C_{(s)} \tag{18}$$

$$CO_2 + 2H_2 \leftrightarrow 2H_2O + C_{(s)} \tag{19}$$

### 3.5. Carrier Steam Distillation

In the petrochemical industry, what is known as carrier steam distillation is usually used to separate hydrocarbon mixtures. Carrier steam distillation constitutes a special case of distillation that enables mixtures to be gently separated. It uses the addition of the vapor pressures of immiscible liquids. The mixture to be separated is evaporated with a low-boiling entrainer (often water) so that the boiling temperature is reduced. The desired fractions can then be drawn off from the carrier steam distillation column via side draws, and the entrainer can then be separated off again [48].

At this point, it should be noted that the petrochemical products are rarely specific chemicals, but are usually mixtures of different components with different properties. To characterize these mixtures and design separation processes, therefore, boiling point ranges or specific temperatures along these boiling curves are generally used. The temperature at which the mixture begins to evaporate is referred to as the initial boiling point (IBP), and the temperature at which the mixture has completely evaporated is called the final boiling point (FBP). The temperature at which a certain volume, for example, 10% of the liquid has evaporated is called the 10% point or T10. More information on the characteristic points is presented by Wegener [43] concerning the boiling curve of jet A-1 aircraft fuel [48].

### 3.6. Technology Readiness Level

In this section, the well-known technology readiness level (TRL) method is used to evaluate the power-to-fuel process developed in this paper. The TRL method indicates the maturity of a technology on a scale from 1 to 9. The method was originally developed by NASA [49] and is now used with adapted definitions in various areas [50]. In this work, the definitions established by the European Commission for the renewable energy sector are employed [51].

The TRLs of $CO_2$ capture technologies range from medium to very high. According to Schmidt et al. [8], for example, $CO_2$ separation from industrial waste gases by means of amine scrubbing (MEA) is already in use and has a TRL of 9. The separation of $CO_2$ from the air, however, is still at an earlier stage of development and assigned a TRL of 6 [8]. As described in Section 3.1, high-temperature electrolysis was used as part of "Fuel 1" by Sunfire GmbH in a larger demonstration plant and therefore assigned a TRL of 5 [13]. High-temperature co-electrolysis has only just started its test phase using the SUNFIRE-SYNLINK technology in 2019, and accordingly has not yet reached the same level of maturity as high-temperature water electrolysis. The remaining technologies used as part of the developed power-to-fuel process are highly developed technologies that are already in use on an industrial scale. The TRLs of these are correspondingly high. There are no specific statements regarding the values for the TRL of carrier steam distillations and reformers, but Luyben [48] describes the industrial use of carrier steam distillations and, it is well known that reformers are used in large-scale processes.

The TRLs of the individual components of the power-to-fuel process considered in this work are, with the exception of high-temperature electrolysis and $CO_2$ separation from the ambient air, very high. However, according to Schmidt et al. [8] and Marchese et al. [33], the TRL of a power-to-fuel process automatically falls to the lowest TRL in the process chain. According to this, the TRL of the developed power-to-fuel process in this work was assessed as 3 due to the low TRL level of the co-electrolysis step.

## 4. Modeling and Simulation in ASPEN PLUS

This section is dedicated to the modeling of the power-to-fuel process in Aspen Plus. For this purpose, the material data and property data models used are first presented. Then, the procedural design of the individual process components as well as the respective selected operating conditions are explained in more detail. For the sake of clarity, the relevant sections of the process flow diagram created in Aspen Plus are shown in Sections 4.2–4.6. When modeling the process, care is taken to ensure that the simulation is suitable for any mass flows.

### 4.1. Material Property Data and Material Property Data Models

As part of the power-to-fuel process developed, a low-temperature Fischer–Tropsch synthesis was used. Hence, with respect to de Klerk [36,52] and Dry [53], only straight-chain, unbranched alkanes with the empirical formula $C_nH_{2n+2}$ were considered. The material data of the hydrocarbons $C_1$ to $C_{29}$ were obtained from the database integrated in Aspen Plus. Based on Schemme [54], the $C_{30+}$-hydrocarbons were viewed as three groups of pseudo-components, with the $C_{30-35}$-, the $C_{36-47}$-, and $C_{48+}$-hydrocarbons grouped together. A representative molecular structure was selected for each of the three pseudo-components. The American Petroleum Institute (API) method, with the data given in Table 1, was used to calculate the thermodynamic properties of the pseudo-components. The Aspen Plus database was used for the material data of components $H_2$, $H_2O$, CO, and $CO_2$. The material data models used were selected on the basis of the general recommendations of Carlson [55] and other application-specific literature. A total of four different material data models were used to simulate the process.

**Table 1.** Properties of the pseudo-components [56].

| | Pseudo-Components | | |
| --- | --- | --- | --- |
| | $C_{30-35}$ | $C_{36-47}$ | $C_{48+}$ |
| Representative molecular structure | $C_{32}H_{66}$ | $C_{41}H_{84}$ | $C_{61}H_{124}$ |
| Molar mass in g/mol | 454.9 | 572.2 | 861.7 |
| Relative density at 60 °F (≈15.6 °C) | 0.818 | 0.827 | 0.839 |
| Boiling point at 1 atm in °C | 469.3 | 528.1 | 624.0 |

The equation of state of Soave–Redlich–Kwong is widely used in the field of gas processing. In the context of this work, the equation of state with reference to Marchese et al. [33] was used as a material data model for the modeled electrolysis types. In order to be able to more precisely calculate the phase equilibrium between gas and liquid in the presence of hydrocarbons and light gases such as $CO_2$ and $H_2$ in the supercritical range, the Soave–Redlich–Kwong equation of state can be extended to the RKS–BM material data model with the Boston–Mathias alpha function. Drawing on Schemme [54], this model was used to calculate the reformer and the parts of the product separation with very low proportions of pseudo-components. The material model Braun K-10 was used to calculate the material flows with higher proportions of pseudo-components. This material data model was especially developed for calculating hydrocarbon mixtures with both real and pseudo-components, and was used in simulations for the Fischer–Tropsch reactor and hydrocracker. The NRTL model enables the description of gas–liquid equilibria as well as liquid–liquid equilibria of strongly non-ideal mixtures. The activity coefficients of the liquid phases are calculated on the basis of experimentally-determined binary interaction parameters. The calculation of CO, $CO_2$, and $H_2$ also takes Henry's law into account. In the simulation, the NRTL–RK material data model was used to calculate the carrier steam distillation. In general, the calculation of the gas phase is by default carried out using ideal gas law. In this work, the Redlich–Kwong equation of state was used instead to describe the gas phase. The NRTL–RK material data model was also used for heat exchangers in which a large proportion of water is in liquid form.

### 4.2. Co-Electrolysis and Water Electrolysis

As Aspen Plus does not have a stored model for the simulation of high-temperature electrolysis, a combination of different Aspen Plus blocks, also known as "Unit Operations", must be used to calculate co- and water electrolysis. In addition, so-called design specs are employed to establish the required process conditions. Similar configurations for simulating high-temperature electrolysis have already been used by Cinti et al. and Marchese et al. [31,33]. The simulation flow diagram for co-electrolysis is shown in Figure 2. It should be noted that not all heat flows of the process can be seen directly in the flow diagram. Various operating resources were used in the simulation to provide the required heating or cooling capacity. For instance, W-5 is an air cooler and W-6 a water cooler, each of which uses the corresponding operating media "air" and "cooling water". The balancing of the resources used was carried out through the "utilities" function integrated in Aspen Plus and is discussed in greater detail in Section 5 The same applies to all of the following flow diagrams.

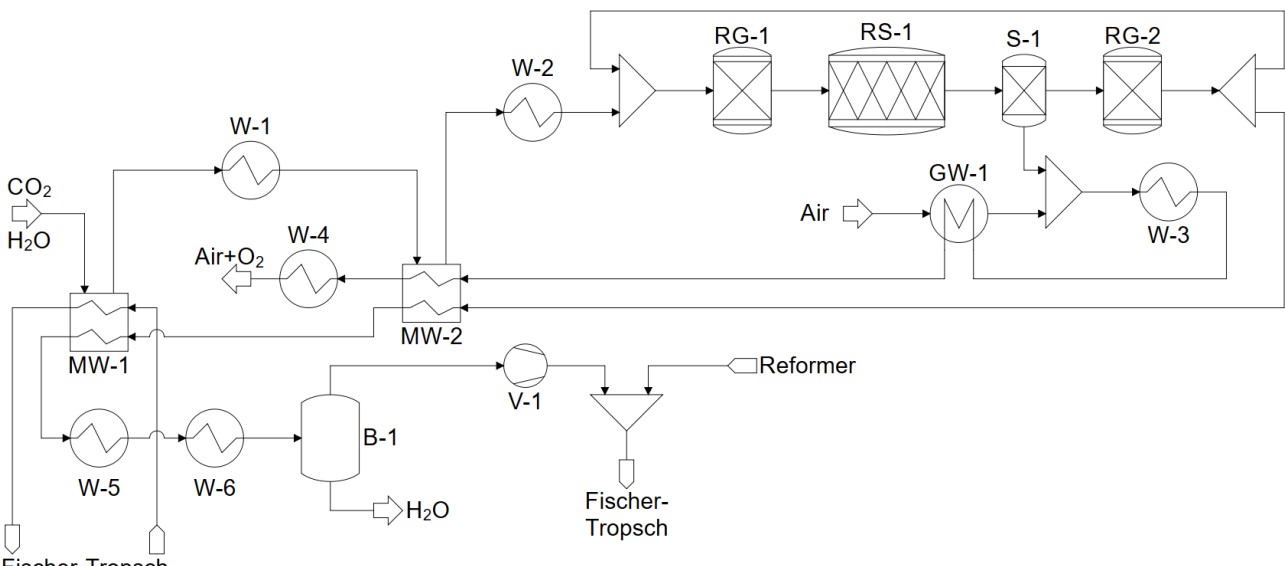

**Figure 2.** Excerpt from the process flow diagram: co-electrolysis.

First, the mixture of $CO_2$ and water, which is present at 1 bar, is warmed up over several stages in a heat exchanger and the water is evaporated. For this purpose, both the waste heat from co-electrolysis and the product flow of the Fischer–Tropsch synthesis as well as part of the waste heat from the Fischer–Tropsch reactor itself, was used. In addition, an electric heater was used with W-2, which ensured that the feed stream reached the electrolysis inlet temperature of 780 °C. The hot gas stream consisting of water vapor and $CO_2$ then enters the electrolysis cell. As already described, the electrolysis cell consisted of several unit operations with which the cathode (RG-1, RS-1, RG-2), the electrolyte (S-1), and anode (GW-1, W-3) were modeled. The feed stream was combined with a recycling stream and passed into a first equilibrium reactor (RG-1). Taking into account the equilibrium reactions that occur (WGS or RWGS: Equation (6), methanation: Equations (8) and (9)), this determines the composition of the gas flow by minimizing the Gibbs energy. Due to the high temperatures, it was assumed that the reaction equilibrium would be reached quickly (see Sun, et al. [57]). The Boudouard reaction (Equation (9)) was not taken into account in the simulation because, according to Sun, Chen, Jensen, Ebbesen, Graves and Mogensen [57], there is no deposition of solid carbon under the selected operating conditions of 800 °C and 1 bar. In the next step, the gas flow passes into the stoichiometric reactor RS-1, in which the electrolysis reactions (Equations (1)–(3)) take place. The conversion of RS-1 was automatically set with a design spec so that the total conversion of the electrolysis cell reactant utilization (RU) corresponded to the specified RU (see Equation (20) [33]). The

total turnover was set at 70% with reference to Sun, Chen, Jensen, Ebbesen, Graves and Mogensen [57] and Marchese, Giglio, Santarelli and Lanzini [33]:

$$RU = \frac{\dot{n}_{react,\,in} - \dot{n}_{react,\,out}}{\dot{n}_{react,\,in}} \qquad (20)$$

The electrolyte was modeled as a simple separator block (S-1), which separates the oxygen produced by the electrolysis reactions. The composition of the synthesis gas was then adjusted in parallel to RG-1 in a further equilibrium reactor (RG-2). While most of the synthesis gas then leaves the electrolysis segment, a portion of the stream is fed back. The size of the returned portion was set using a design spec so that the $H_2$ concentration in the feed of the electrolysis was at least 10 mol% in order to avoid oxidation of the nickel-based cathode [27,58]. Synthesis gas leaving the electrolysis cell is gradually cooled down for improved energetic utilization, and the unconverted water is condensed out. The synthesis gas is then compressed to 30 bar by a multi-stage compressor and merged with the reformed synthesis gas (see Section 4.6). Thereby, the $H_2/CO$ ratio required for the Fischer–Tropsch synthesis was available after mixing of the synthesis gas streams and a further design spec was used that adjusts the ratio of water to $CO_2$ in the feed stream of the co-electrolysis system accordingly.

MW-1 and MW-2 represent multi-component counter-flow heat exchangers, which were used to recover the process heat for educt conditioning. W-1 is a heater used for further educt conditioning. V-1 represents a compressor. The separator block B-1 is used to separate the vapor phase and the liquid phase at equilibrium.

The oxygen stream separated by the separator block functioning as an electrolyte was mixed with an air stream on the anode side of the electrolytic cell. According to Cinti et al. [31], it is common practice to carry the oxygen through a stream of air in order to prevent the cell performance from being negatively influenced by a high concentration of the oxygen. The amount of air flow was set by a design spec so that the partial pressure of the oxygen after mixing with the air was 0.5 bar [31]. The air flow was warmed up as much as possible using a counter flow heat exchanger (GW-1) before it entered the electrolysis cell. As the cell temperature of 800 °C was not reached as a result, a heat exchanger block (W-3) was used to take into account the additional heating output that must be provided by the electrolysis cell. The air flow was then cooled in two steps and left the system air-enriched with oxygen.

A simplified flow chart of the water electrolysis for hydrogen provision for the hydrocracker is displayed in Figure 3. Its structure corresponded to that of co-electrolysis, but with the simplification that no equilibrium reactors were used. These were not required, as there was no $CO_2$ or CO present in the water electrolysis segment. Another difference is that the hydrogen was compressed to the operating pressure of the hydrocracker of 50 bar. Using the same logic as in the previous figure, W-7 and W-8 were heaters for steam generation, assisting the multi-component counter-flow heat exchanger MW-3. RS-3 was the stoichiometric reactor representing the water electrolysis. The separator block S-2 divided the products to anode and cathode sections. W-9 and W-10 were used to cool the product mixtures of the water electrolysis, GW-2 was a counter-flow heat exchanger used for pre-heating air feed, and V-2 was used to compress the produced hydrogen.

For the techno-economic analysis of the power-to-fuel process, it is necessary to determine the performance of the co- and water-electrolysis processes. For this purpose, the calorific value of the product flows could be output via Aspen Plus and the electrolysis output could then be calculated using a specified efficiency (see Equation (10)).

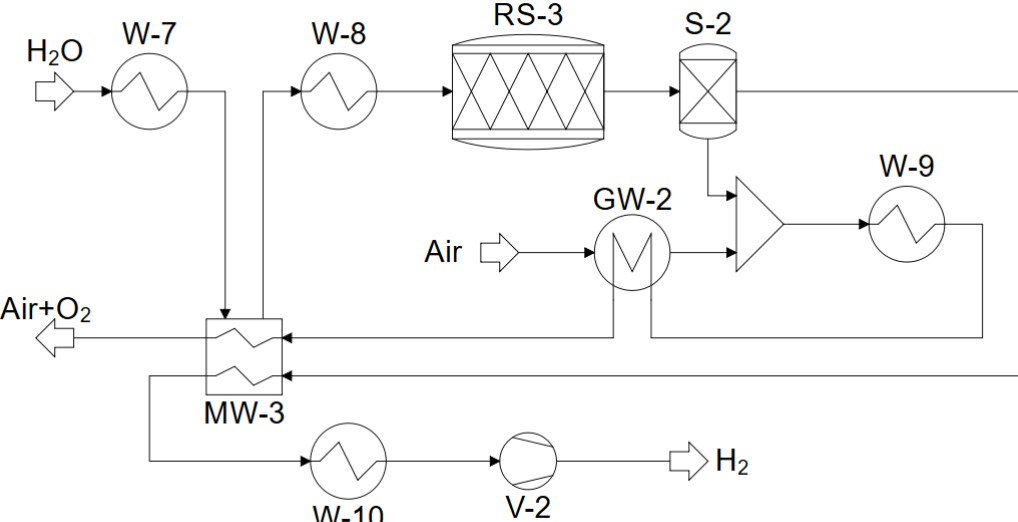

**Figure 3.** Excerpt from the process flow diagram: water electrolysis.

### 4.3. Fischer–Tropsch Synthesis and Product Separation

The flow chart of the Fischer–Tropsch synthesis and the subsequent product separation is shown in Figure 4. The synthesis gas from the co-electrolysis was mixed with the synthesis gas from the reformer and fed into the Fischer–Tropsch reactor (RS-2). This was modeled as an isothermal bubble column reactor with a pressure of 30 bar and a temperature of 210 °C. A stoichiometric reactor was used for modeling, in which the reaction equations for 32 parallel reactions according to Equation (11) were stored (for $n = 1$ to $n = 29$ and the pseudo components corresponded to $n = 32$, $n = 41$ and $n = 61$). The conversions of the various reactions were determined using a calculator block with an integrated Excel file, based on the ASF distribution (Equation (12)), with the chain growth probability $\alpha$ calculated using Equation (14). The modeling was therefore suitable for calculations with variable $H_2/CO$ ratios; however, a ratio of 1.8 was chosen in the context of this work in order to maximize the proportion of middle distillates in the product of the Fischer–Tropsch synthesis. Under the selected operating conditions and the set $H_2/CO$ ratio, there was a chain growth probability of approximately 0.92. According to de Klerk [36], this value is in the range of typical industrial low-temperature Fischer–Tropsch syntheses. The total conversion of carbon monoxide was set at 80%, reflecting the work of Becker, et al. [59], Trippe [35], and Schemme [54].

As the stoichiometric reactor used only had one product stream, in a first step (B-2) the product from the Fischer–Tropsch reactor was divided isothermally and isobarically into gas and liquid phases. The liquid phase, which consisted mainly of $C_{20+}$-hydrocarbons, was fed to the hydrocracker, while the gas phase was cooled isobarically in several steps to 40 °C, with part of the waste heat being used in the co-electrolysis process. After cooling, the light gases were separated from the middle distillates in the next container (B-3). Almost all of the $C_{8+}$ hydrocarbons that could be used for kerosene and diesel were sent to the carrier steam distillation, and the short-chain hydrocarbons were sent to the reformer. In addition, the differences in density and polarity of the water and hydrocarbons in both tanks were used to separate the water produced by the Fischer–Tropsch synthesis. In order to avoid soot formation in the reformer, some of the separated water was fed into the reformer. The proportion fed into the reformer was determined with a design spec corresponding to the selected $H_2O/C$ ratio (see Section 4.6).

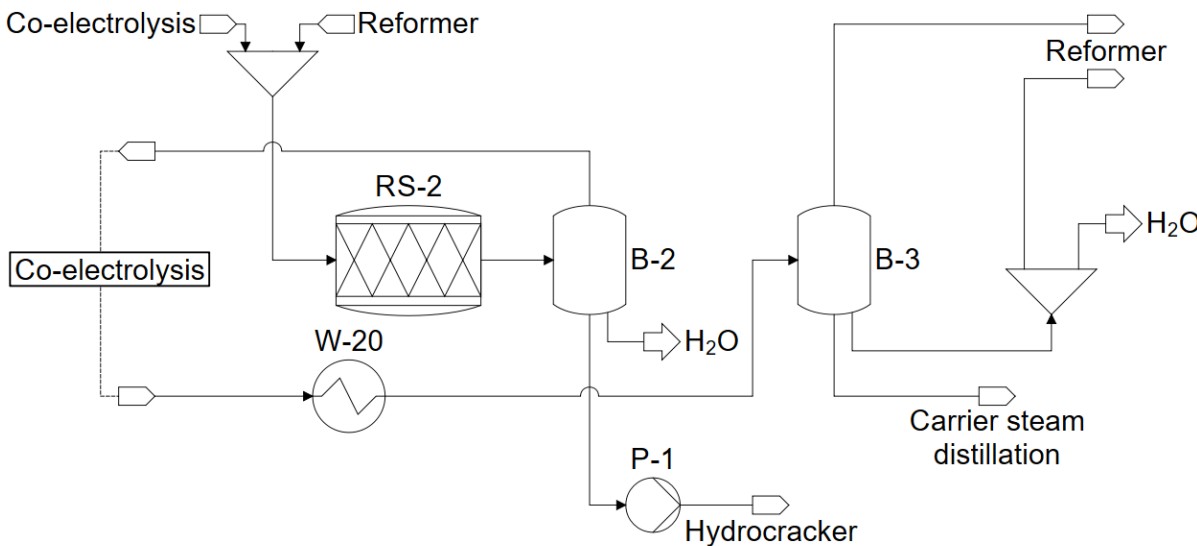

**Figure 4.** Excerpt from the process flow diagram: Fischer–Tropsch synthesis and product separation.

*4.4. Hydrocracker*

Figure 5 displays the modeled hydrocracker. The long-chain hydrocarbons were brought together with several re-circulations in which the required hydrogen was also added. The hydrogen was produced using water electrolysis and the total amount adjusted so that the mass fraction of hydrogen after mixing was around 8% (mass). This value was chosen as a compromise between the suppression of both soot formation and catalyst deactivation (see Section 3.3) and minimization of the required additional electrolysis performance.

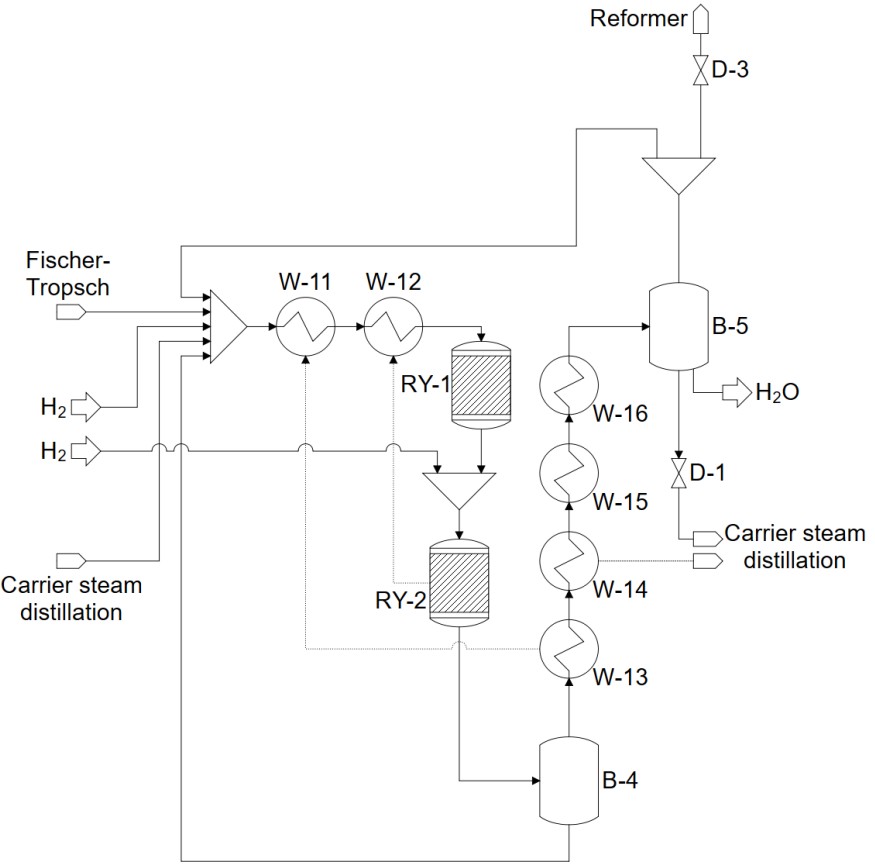

**Figure 5.** Excerpt from the process flow diagram: hydrocracker.

The mixed material flow was preheated and fed into the hydrocracker. The hydrocracker was modeled as a two-stage isothermal fixed bed reactor, whereby the two fixed beds were each modeled by a RYield block (i.e., a yield reactor). To ensure ideal hydrocracking, an operating pressure of 50 bar and a temperature of 350 °C were selected and a conversion per bed of 60% assumed. The yields of the respective cracking products were calculated in calculator blocks connected to Excel files, based on the uniform distribution of the cracking products of the ideal hydrocracking, which is described in more detail in Section 3.3 The heat required for the cracking process was provided by the exothermic reformer (see Section 4.6). The product stream was then separated into gas and liquid phases, and the long-chain hydrocarbons that had not been cracked were returned to increase the conversion of the hydrocracker. The gas stream was cooled in several stages for better energy utilization and the middle distillates were separated off, expanded to 30 bar, and passed to the carrier steam distillation. The gas stream separated in B-5 consisted mainly of hydrogen and $C_{7-}$-hydrocarbons, $H_2O$ and $CO_2$, and was partly returned to the hydrocracker in order to reduce the need for fresh hydrogen and thus the required performance of the water electrolysis system. A purge flow of 10% was expanded to 30 bar and fed to the reformer to prevent $CO_2$ and short-chain hydrocarbons from accumulating in the hydrocracker.

*4.5. Carrier Steam Distillation*

The feed streams of the carrier steam distillation, which can be seen in Figure 6 were the product stream of the hydrocracker unit and the separated middle distillates from the product separation following the Fischer–Tropsch synthesis (see Figure 4). As the NRTL–RK material data model was to be used in the distillation column, which is not suitable for calculating mixtures with pseudo components, the remaining pseudo components (approximately 0.6% by weight) were assigned to the real component $C_{29}H_{60}$. A RYield block (RY-3) was used for this. In the next step, the feed was directed into a so-called pre-flash drum (B-6), in which part of the material flow was evaporated by releasing it to 1 bar. The use of a pre-flash drum is recommended in the literature [48] for the separation of some of the gases and thus simplify product separation in the column. In the simulation, almost all of the remaining $CO_2$, hydrogen, and methane as well as most of the $C_2$- to $C_5$-hydrocarbons were separated by the flash evaporation. The liquid phase was preheated by heat integration and fed to the vaporizer (furnace) of the column, which was modeled with a Petrofrac block (K-1). The energy required to evaporate the feed was provided by the exothermic reformer (see Section 4.6). Steam was fed to the column via the sump, which reduced the partial pressure of the hydrocarbons and thus lowered the boiling temperatures. The column had two strippers with four stages and their own steam supplies, each for kerosene and diesel. There was no need for a side exhaust for gasoline, which is common in industrial applications. The number of stages in the column as well as the position of the feed and the side draws were determined using a sensitivity analysis in such a way that as much kerosene and diesel as possible was produced, but the requirements of the standards (see Table 2 were still met. The requirements of the standards were taken into account in two design specs through which the specified T50 temperature of the respective product flow was set by automatically varying the amount of kerosene or diesel withdrawn. At this point, it should also be noted that the use of pump-arounds was dispensed with. Pump-arounds, through the recirculation of material flows within the column, represent a possibility for recovering thermal energy and reducing the column's energy requirement [48]. In the simulation, however, the use of pump-arounds led to considerable convergence problems. The bottom product, consisting of long-chain hydrocarbons, was returned to the hydrocracker, and the top product, together with the gas flow from the pre-flash drum, was compressed to 30 bar and sent to the reformer. A detailed analysis of the fuels produced is presented in Section 5.1.

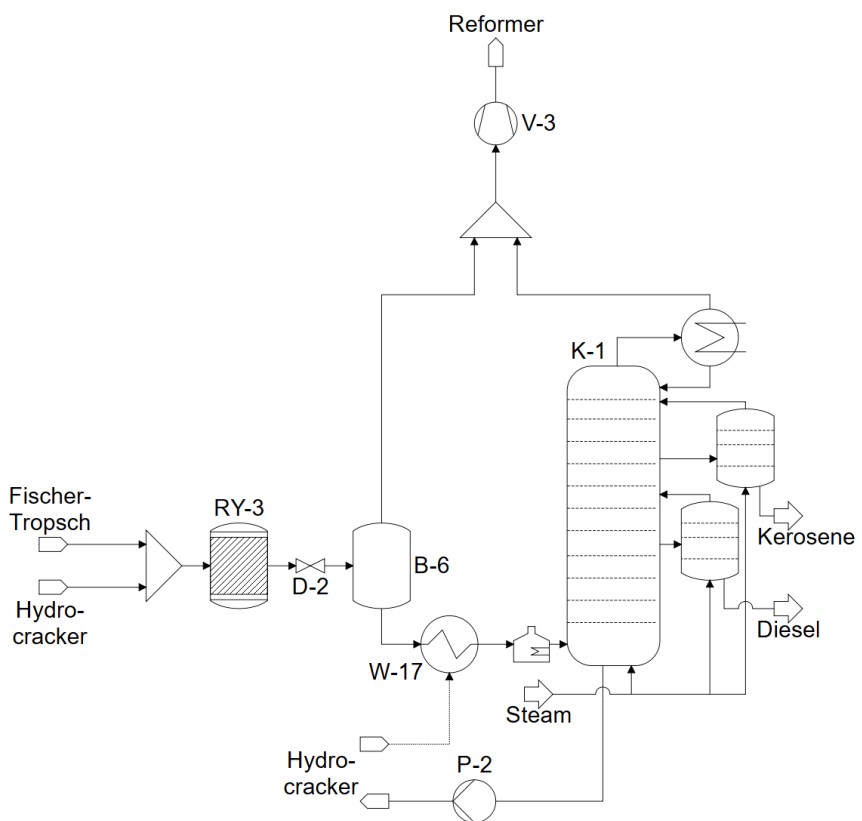

**Figure 6.** Excerpt from the process flow diagram: carrier steam distillation.

**Table 2.** Characteristic values of the synthetic kerosene and diesel produced.

| Characteristic Value | Unit | Kerosene FT-SPK (ASTM 7566) | Syn. Kerosene (Simulation) | Diesel Class A (EN 15940) | Syn. Diesel (Simulation) |
|---|---|---|---|---|---|
| T10 | °C | ≤205 | 157 | - | - |
| T95 | °C | - | - | ≤360 | 356 |
| T90–T10 | K | ≥22 | 70 | - | - |
| FBP | °C | ≤300 | 274 | - | - |
| Density @ 15 °C | kg/m$^3$ | 730–770 | 738 | 765–800 | 779 |
| Cetane number | - | - | - | ≥70 | 120 [1] |
| Freezing point | °C | ≤−40 | n/a | - | - |
| Heating value (LHV) | MJ/kg | - | 44.17 | - | 43.85 |

[1] Probably overrated; for an explanation see text.

### 4.6. Reformer

The reformer was used to convert the unusable gases into synthesis gas and thus to recycle them. The corresponding section of the simulation flow diagram is shown in Figure 7. The gas streams available at 30 bar from the distillation column, the hydrocracker and the product separation were combined with the addition of water. The amount of water added was automatically set using a design spec so that there was an $H_2O$/C ratio of 0.6 after mixing. The feed stream was then heated to 900 °C with the waste heat from the product stream and passed into the reformer. The reformer was modeled with an RGibbs block (i.e., an equilibrium reactor). Due to the high operating temperature, the short residence time and gas phase reaction, the kinetics can be neglected [54] and the product composition can be determined by minimizing the Gibbs energy. The addition of oxygen was automatically controlled by a design spec so that a product temperature of 950 °C results. The reformer was exothermically-operated in order to provide the high-temperature heat required for the hydrocracker and the distillation column through the waste heat. The additional heat to be

produced was taken into account by a cooling capacity imposed on the reformer from the outside. The reformed synthesis gas was cooled and then mixed with the synthesis gas from the co-electrolysis unit and fed into the Fischer–Tropsch reactor.

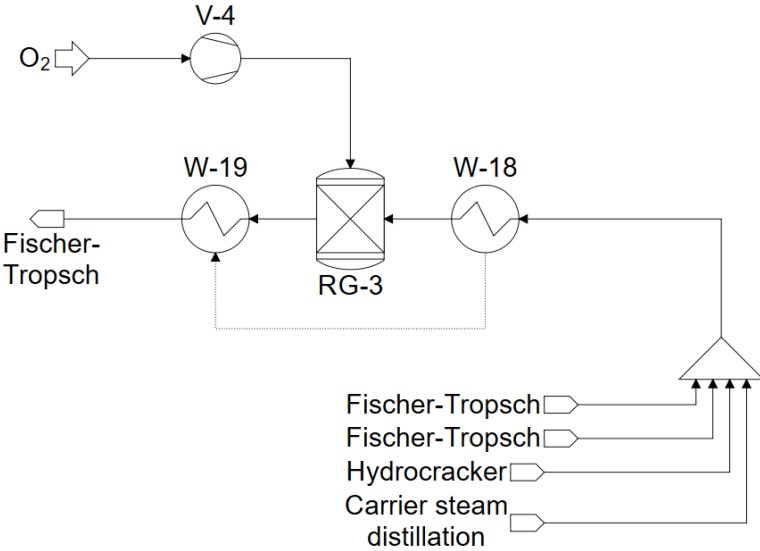

**Figure 7.** Excerpt from the process flow diagram: reformer.

## 5. Results from Process Analysis

In this section, the simulation results of the power-to-fuel process presented in Sections 3 and 4 are presented. First, it was determined whether the diesel and kerosene fuels produced met the requirements of the relevant standard. The composition of the fuels was considered, the boiling curves were calculated and compared with the real boiling curves of kerosene and diesel, and finally the fuel property data were checked. Then, the process balance was considered and the PtL efficiency of the process established. The next step was to examine the extent to which heat extraction for the provision of thermal energy for $CO_2$ separation is possible and useful. Finally, the developed process was compared with other power-to-fuel or PtL processes.

### 5.1. Fuel Property Analysis

The mass fractions of the various carbon chains in the kerosene and diesel produced as well as the mass-related average molecules of the respective fuels are shown in Figure 8. The kerosene withdrawn from the carrier steam distillation column contained almost no $C_1$- to $C_7$-hydrocarbons and approximately 8.3% by weight of $C_8$-hydrocarbons. The mass fraction of $C_9$ to $C_{11}$ is between 18 and 19% by weight and the fraction of $C_{12}$ is around 16.5% by weight. The proportion of longer hydrocarbons steadily decreased, with $C_{13}$ being contained in kerosene at approximately 11.6% by weight, $C_{14}$ at approximately 5.4% by weight, and $C_{15}$ at approximately 1.8% by weight. Carbons with a chain length of 16 and longer are only contained in kerosene to a very small extent, with a total of around 0.7% by weight. The mass average chain length of the hydrocarbons in kerosene is 10.9. The diesel fuel produced consisted of approximately 0.7% by weight of carbons with a chain length of eleven or less. The mass fraction of longer hydrocarbon chains increased steadily from $C_{12}$ at about 2% by weight, to $C_{15}$ at 11.5% by weight. The curve of the mass fractions then flattened out and the $C_{16}$- to $C_{20}$-hydrocarbons were each contained in the diesel fuel at approximately 12.2% by weight. The mass fraction of longer hydrocarbons initially dropped sharply at 2.1% by weight for $C_{21}$, and then dropped uniformly with increasing chain lengths to approximately 0.3% by weight of $C_{28}$ hydrocarbons. The proportion of $C_{29}$-hydrocarbons was around 1.2% by weight. The mass-related average chain length of diesel fuel was 17.5.

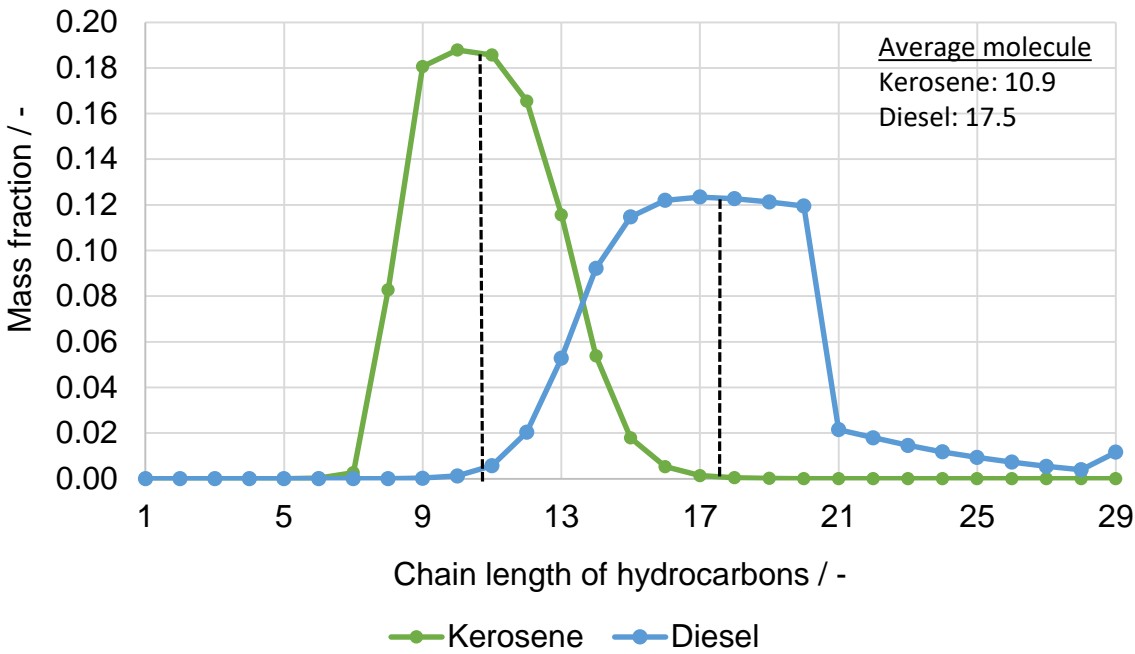

**Figure 8.** Product distribution of the produced kerosene and diesel fraction.

In the literature, chain lengths of between 8 and 16 carbon atoms were assigned to the kerosene fraction [37]. Carbons with these chain lengths made up more than 99.5% by weight of the kerosene produced in the simulation. The product distribution of synthetic kerosene was therefore in the range of typical aircraft fuels. The sharp drop in the mass fractions from $C_{20}$ to $C_{21}$ in the diesel produced can be explained by the fact that the proportion of $C_{21+}$-hydrocarbons in the feed of the distillation column was also low. This was, in turn, due to the majority of the $C_{21+}$ hydrocarbons produced in the Fischer–Tropsch reactor being fed directly to the hydrocracker and cracked there. The comparatively high proportion of $C_{29}$ can be attributed to the fact that, as described in Section 4.4, the pseudo-components were combined with the $C_{29}$ hydrocarbons. Therefore, the 1.2 wt.% not only consisted of $C_{29}$ hydrocarbons, but also contained all of the hydrocarbons with a chain length of 30 or more. Which hydrocarbon chains are assigned to diesel varies with the literature source. For example, Trippe [35] only counted the $C_{13}$ to $C_{20}$ hydrocarbons in the diesel fraction, whereas Bacha, et al. [60] and Dieterich et al. [6] assigned hydrocarbons with chain lengths of 10 to 22 or 23 to the diesel fraction. The synthetic diesel fuel produced in the simulated power-to-fuel process consisted of approximately 86.8% by weight of $C_{13}$ to $C_{20}$ and approximately 95% by weight of $C_{13}$ to $C_{23}$. Accordingly, the product composition of the diesel produced fell within the range of typical diesel fuels. Both the product composition of the synthetic kerosene and that of the synthetic diesel were therefore within the acceptable range.

In the next step, the boiling curves of the fuels produced in the simulation were calculated and compared with the real boiling curves of kerosene Jet A-1 and diesel (see Figures 9 and 10). The boiling curves were calculated using the D86CRV method integrated in Aspen Plus, which calculates the boiling curve of a mixture of substances at atmospheric pressure.

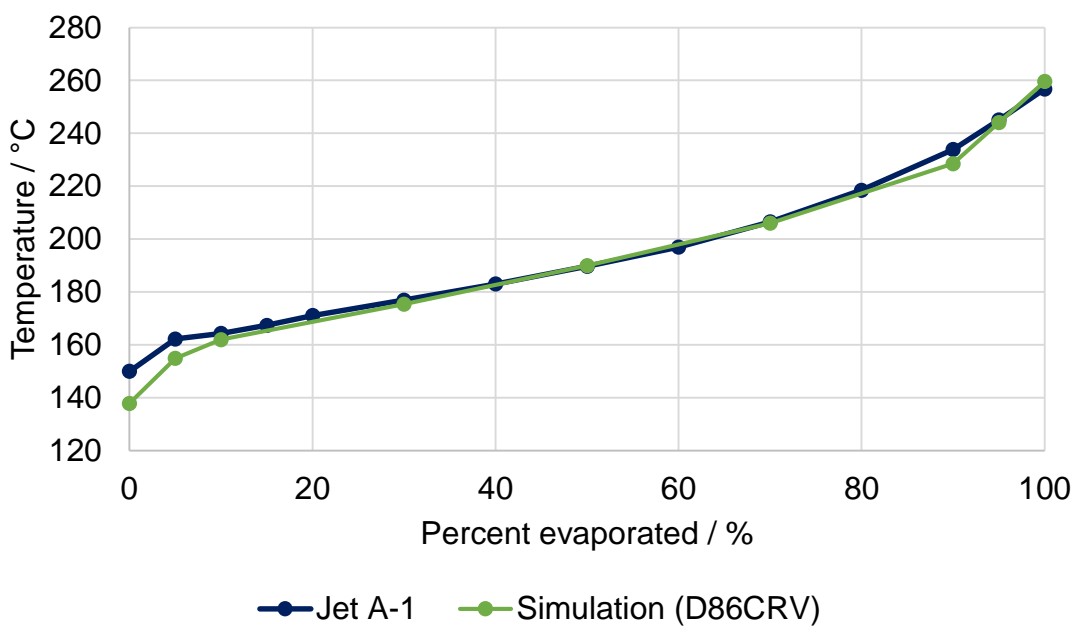

**Figure 9.** Boiling curves of jet fuel in comparison to Jet A-1 (boiling curve according to Edwards [61]).

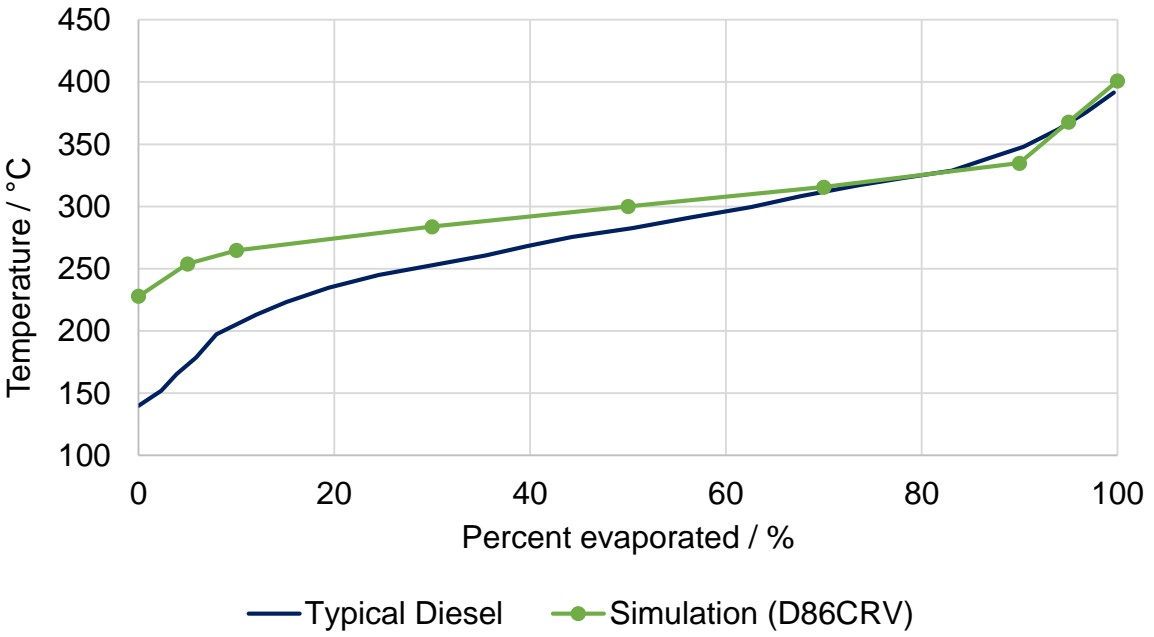

**Figure 10.** Boiling curves of diesel in comparison (boiling curve of "typical diesel fuel" according to Bacha et al. [4]).

The calculated boiling curve of synthetic kerosene and that of Jet A-1, according to Edwards [61], are shown in Figure 9. Both curves were almost identical, with the most noticeable difference being that the IBP of the synthetic kerosene of the simulation (137 °C) was below that of the reference kerosene of type Jet A-1 of approximately 150 °C. The T5 points of the boiling curves deviated by approximately 7 K at 155 °C for the calculated value and approximately 162 °C for the reference. With the exception of the T90 point, the boiling curves deviated from each other by less than 3 K in the area from T10 to the FBP, the deviation at the T90 point being approximately 5 K. Overall, it can be concluded that the boiling behavior of the kerosene produced in the simulated power-to-fuel process corresponded well to that of jet A-1 kerosene.

Figure 10 shows the calculated boiling curve of the synthetic diesel fuel produced and the boiling curve of a "typical" diesel fuel according to Bacha et al. [60]. A comparatively large difference could be seen between the IBP of the two boiling curves. The IBP of the reference diesel was around 140 °C, whereas that of the diesel fuel produced was around 228 °C; about 88 K higher. The boiling curves converged as the proportion of evaporated volume increased. The T10 points were about 55 K apart at about 210 °C for the reference diesel and 265 °C for the calculated boiling curve, whereas the T50 points differed by about 16 K at approximately 283 °C and 299 °C.

As the proportion of evaporated fuel increased, the boiling curves further converged. The FBP of the reference diesel was around 390 °C and that of the calculated boiling curve was around 400 °C. Overall, the deviations between the boiling curves of the reference and synthetic diesels were greater than those between the boiling curves of kerosene. Larger deviations could be found, especially in the area spanning the IBP to the T50 point. When comparing the compositions of the two diesel fuels, however, it was noticeable that the reference diesel according to Bacha et al. [60] had a comparatively higher proportion of short-chain light hydrocarbons. The average chain length of the hydrocarbons of the reference diesel was approximately 16, and accordingly, below the average chain length of the diesel fuel produced of 17.9. The deviations in the boiling curve can be explained by the correspondingly different compositions of diesel fuels. In order to determine whether the kerosene and diesel fraction produced is suitable for use as fuels, the requirements of the respective standards were checked in the next step. Table 2 compares the requirements of the standards and corresponding characteristic values of the fuels calculated in the simulation. In addition, the calorific values of synthetic kerosene and diesel are given, which were calculated using the Aspen Plus component properties (property sets) QVALNET.

The T10, T95, and T90 temperatures as well as the FBP of the synthetic kerosene were read from the boiling curves and were in the required range for both the synthetic kerosene and synthetic diesel. The densities of the two fuels were determined by the Aspen Plus component property RHOLSTD. The density of the FT SPK (738 kg/m$^3$) was at the lower end of the permitted range, but still met the requirements of ASTM 7566. The density of the synthetic diesel was also within the required range at 779 kg/m$^3$. The Aspen Plus component property CETANENO was used to determine the cetane number of synthetic diesel. This yielded a cetane number of 109.2 for pure n-hexadecane ($C_{16}H_{34}$). According to Dry [53], however, the actual cetane number of n-hexadecane is 100. Accordingly, it can be assumed that the actual cetane number of synthetic diesel is below the calculated value of 120. The freezing point of kerosene cannot be calculated using Aspen Plus. Due to the very high proportion of n-alkanes in the product of the Fischer–Tropsch synthesis [9], and therefore a very high proportion of n-alkanes in kerosene, there is the possibility that an after-treatment of the kerosene is necessary in order to improve the low-temperature properties. One possibility would be isomerization, as isoalkanes have a significantly lower freezing point [62]. On the basis of this sub-section, it should be noted that both of the synthetic fuels produced, namely kerosene and diesel, met the requirements of the respective standards. However, if the synthetic kerosene is to be used as a mixture component for Jet A-1, an after-treatment may be necessary to improve the low-temperature properties.

The following section deals with the balance of the process and presents the amount of energy and material required to produce the synthetic fuels.

### 5.2. Balancing the Power-to-Fuel Process

The simulation of the PtF process created in Aspen Plus is essentially suitable for calculating any mass flows. The process balance based on the production output is discussed in the following section. For this purpose, the unit liter diesel equivalent $l_{DE}$ was defined as 35.9 MJ. First, the materials required for the production of synthetic fuels are discussed. Then, the process balance is considered from an energetic point of view, whereby the balance of the equipment used is also presented. The material balance is shown in Figure 11

as an overview. It should be noted that not all process-internal heat flows are shown in the figure. The heat flows shown in Figure 11 are those that have a major influence on the overall process, and are discussed in more detail below. If 1 $l_{DE}$ is produced in the simulation, this liter consists of 38.9% synthetic kerosene and 61.1% synthetic diesel. A total of 3.99 kg of water per liter of diesel equivalent produced is required as a feed for the water and co-electrolysis as a material for the production of the fuels, of which approximately 0.18 kg of water is used for pure water electrolysis. In addition, $CO_2$ is required for the co-electrolysis process. The required amount is approximately 2.54 kg $CO_2/l_{DE}$. In addition to water and $CO_2$, the power-to-fuel process requires oxygen to operate the reformer. For each liter of diesel equivalent produced, around 0.34 kg of oxygen is required. Due to internal returns and recycling streams in the process, the only waste streams that arise are oxygen-enriched air streams from electrolysis and separated water. An overview of the material balance of the process is presented in Table A1 (Appendix B). In the developed power-to-fuel process, there are various heat sinks and heat sources. In order to minimize the energy requirement, an energy integration was carried out. Heat sinks and sources were partially coupled to each other via direct heat exchange between material flows, and the required heating or cooling capacity was partially provided by operating equipment.

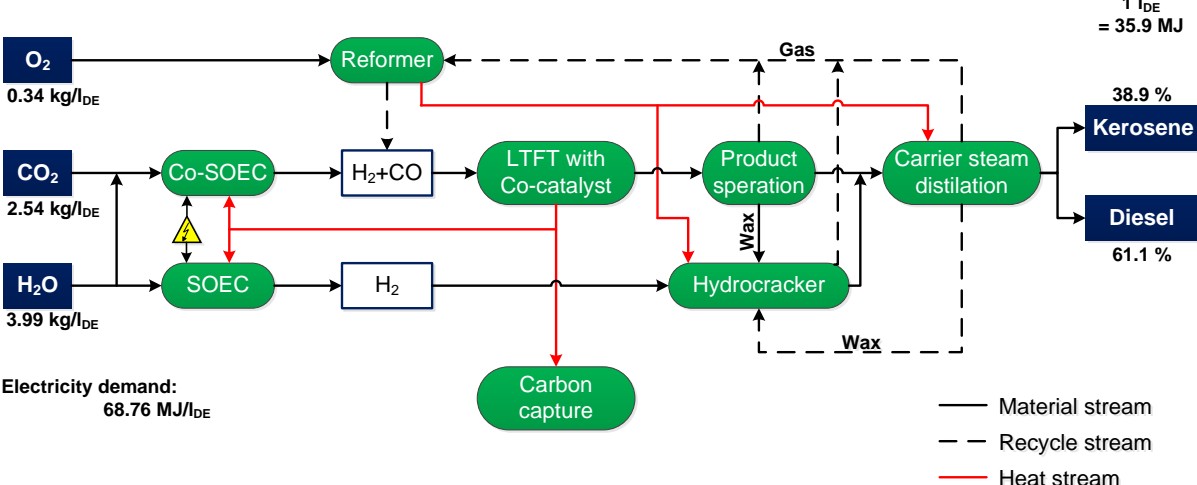

**Figure 11.** Energy-specific material balance of the developed fuel synthesis.

The low- and medium-pressure saturated steam were used as both a heating medium and coolant by generating the corresponding steam. For example, the Fischer–Tropsch reactor was cooled by generating medium-pressure saturated steam with the waste heat from the reaction. In addition, it was assumed that low-pressure saturated steam was used as an entrainer for the operation of the carrier steam distillation column, and this consumption was correspondingly taken into account in the balance. Electricity is required to operate the compressors and pumps as well as the two electrical heaters W-2 (see Figure 2) and W-8 (see Figure 3). An isentropic efficiency of 76% was assumed for the compressors and of 60% for the pumps. The electricity demand for water and co-electrolysis was considered separately.

Table A3 in Appendix B shows that the demand for both low-pressure and medium-pressure steam can not only be covered within the process, but there is even usable thermal energy left over. As displayed in Figure 11, this thermal energy can be discharged from the process and used, for instance, for carbon capture. This heat coupling is examined in greater detail in Section 6.3. The table does not include the heating power required to operate the carrier steam distillation column and the hydrocracker of 0.39 MJ/$I_{DE}$ and 0.41 MJ/$I_{DE}$, respectively. As both the distillation column and hydrocracker are operated at over 300 °C, it is not possible to provide this heat demand using low- or medium-pressure steam. In order that no valuable electrical energy must be used to provide heating power,

the reformer was operated exothermically instead of autothermally, as is common in many industrial applications [46]. The heat made available in this way can, as Figure 11 indicates, be used to operate the hydrocracker and distillation column. Thermal oils can be used for heat transfer as they enable heat transfer at temperatures of up to 400 °C [63]. As a result, with the exception of the high-temperature heat required to operate the water and co-electrolysis, all of the required process heat can be provided via internal heat integration. As described in Section 3.1, the electrical power required for the electrolysis can be calculated using the calorific value of the electrolysis products and determining the efficiency. The calorific value and output of the products can, in parallel to the calorific value of synthetic fuels, be determined using the Aspen Plus component property, QVALNET. For the electrical efficiency of high-temperature electrolysis, values from 60% to over 100% can be found in the literature, depending on the mode of operation of the SOEC and how the balance space of the electrolysis system is selected [64]. If, for example, the required heat is not included in the calculation, if this is potentially available free of charge as waste heat from another process, the efficiency of the electrolysis system improves accordingly. In the developed power-to-fuel process, the low-temperature heat for the evaporation of the water is provided through heat integration and therefore does not need to be taken into account when determining the electrolysis efficiency. However, the required high-temperature heat cannot be provided through heat integration but must be supplied to the system in the form of electrical energy. Therefore, it must be taken into account in the efficiency calculation. In this case, Peters et al. [64] specified an electrolysis efficiency $\eta_{SOEC}$ of approximately 80% for approximately thermo-neutral operation. At this degree of efficiency, however, compression work for storing the electrolysis products is also included. Although this type of compression was not carried out in the developed power-to-fuel process, an electrolysis efficiency of 80% was selected to be on the safe side, in order not to underestimate the required electrical power of the electrolyzers. The amount of synthesis gas or hydrogen calculated in Aspen Plus and the specified efficiency result in the electrical energy required to operate the electrolyzers of approximately 3.09 MJ/l$_{DE}$ for water electrolysis and 58.78 MJ/l$_{DE}$ for co-electrolysis. One of the greatest advantages of energy integration, which is made possible by a combination of high-temperature electrolysis and Fischer–Tropsch synthesis, can also be made clear on the basis of these values. As shown in simplified form in Figure 12, the feed stream of the co-electrolysis is evaporated and preheated in a heat exchanger section (see Figure 2). About 16.4 MJ/l$_{DE}$ of thermal energy was added to the feed stream. For this purpose, the waste heat from the Fischer–Tropsch synthesis was used, amongst other things, via the energy integration. Without energy integration, this amount of energy would need to be added to the high-temperature electrolysis. Conversely, this means that the energy requirement for co-electrolysis is reduced by more than 20% through the energy integration, and that excess thermal energy is also available and can be used for $CO_2$ separation (see Section 5.3).

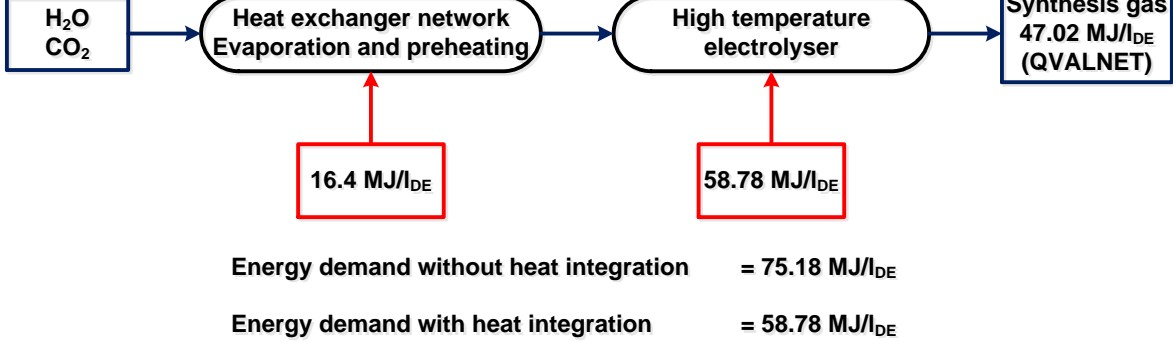

**Figure 12.** Simplified balance for co-electrolysis.

Once the power-to-fuel process has been fully balanced, the overall efficiency of the process, referred to as the PtF or PtL efficiency, can be determined. As the fuel synthesis does not need to be supplied with external thermal energy, the efficiency is calculated on the basis of the production output and electrical energy requirement. The electrical energy requirement was made up of the energy requirement of 58.78 MJ/$l_{DE}$ for co-electrolysis, the energy requirement of 3.09 MJ/$l_{DE}$ for water electrolysis, and the energy requirement for operating all other system components of 6.89 MJ/$l_{DE}$ (see Table A3). It should be noted that $CO_2$ separation is not yet taken into account at this point. The following section deals with the influence of $CO_2$ separation on the PtL efficiency. This results in a power-to-fuel efficiency for fuel synthesis of:

$$\eta_{PTL} = \frac{35.9 \frac{MJ}{l_{DE}}}{6.89 \frac{MJ}{l_{DE}} + 3.09 \frac{MJ}{l_{DE}} + 58.78 \frac{MJ}{l_{DE}}} = 52.21\% \tag{21}$$

It becomes clear that the electrical power required for high-temperature electrolysis and so the electrolysis efficiency exerts the greatest influence on the power-to-fuel efficiency. To take a closer look at this effect, Figure 13 displays the overall efficiency of the process versus that of high-temperature electrolysis. The overall efficiency of the process is linearly dependent on the electrolysis efficiency. For an $\eta_{SOEC}$ of 60%, the $\eta_{PTL}$ drops to 40%. However, according to Peters et al. [64], electrolysis efficiencies of less than 70% only occur when the low-temperature heat for evaporation of the water must be provided by electrical energy. As this heat is available in the developed power-to-fuel process via the energy integration, an electrolysis efficiency of 70% was assumed for the worst case. Accordingly, the lowest possible power-to-fuel efficiency was around 46.26%. In the event that high-temperature heat is available and correspondingly does not need to be provided by electrical energy, electrolysis efficiencies of 100% and higher are possible. If the developed power-to-fuel process is to be built, for instance, in a network location where such high-temperature heat is available, a power-to-fuel efficiency of 63.67% is theoretically possible with an electrolysis efficiency of 100%. At this point, it should be noted that the assumed lower limit of the efficiency of 70% and also the 80% chosen for the base case represent conservative assumptions. Due to the energy integration carried out and the fact that in the developed fuel synthesis no compression of the electrolysis products for storage was carried out, the electrolysis efficiency tended to be in the range above 80%, rather than in that from 70% to 80% [64].

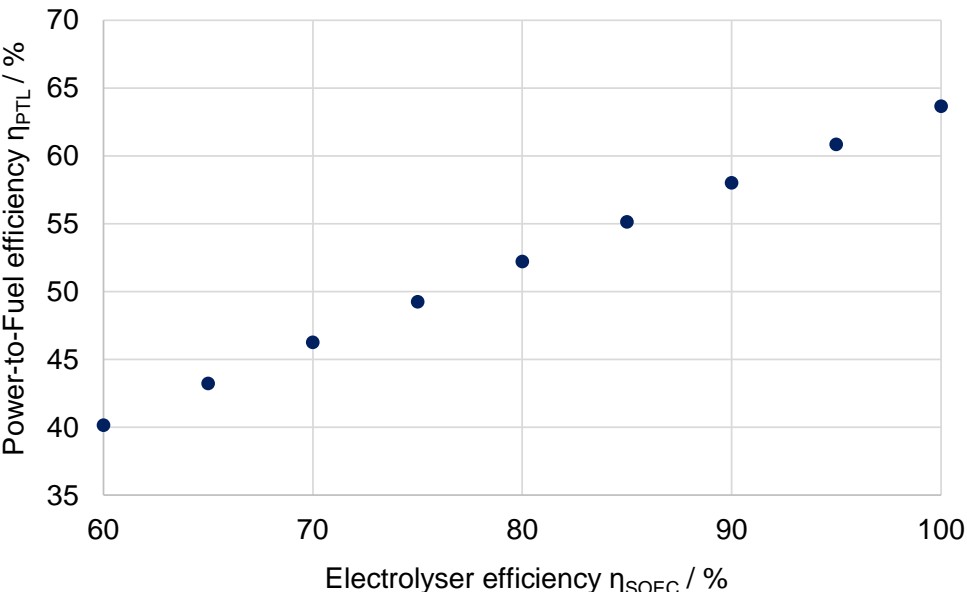

**Figure 13.** Power-to-fuel efficiency as a function of electrolysis efficiency.

## 5.3. Heat Recuperation and $CO_2$ Separation

The possibility of energy integration is often mentioned in the literature as a great advantage of PtL processes based on Fischer–Tropsch synthesis. Energy integration has already been carried out for the developed process in order to provide the required process heat. However, the values listed in Table A3 in Appendix B demonstrate that there are still large amounts of excess heat available that can be removed from the process in the form of low- and medium-pressure steam. In the following, it should be determined whether this heat can be used for $CO_2$ separation in order to provide the $CO_2$ required for co-electrolysis. In addition, the influence of such a process coupling on the efficiency of the entire process chain from $CO_2$ separation to synthetic fuel was examined. For the separation of $CO_2$ from biogases and from industrial exhaust gases, heat is required at a temperature level [65,66] that exceeds that of the excess low-pressure steam. Therefore, only the excess medium-pressure steam is considered in the following. For every liter of diesel-equivalent produced, the power-to-fuel process produces 9.191 MJ of medium-pressure steam. This energy can then be converted into the amount of energy released per unit of $CO_2$ consumed.

$$\frac{9.191 \frac{MJ}{l_{DE}}}{2.54 \frac{kg_{CO2}}{l_{DE}}} = 3.619 \frac{MJ}{kg_{CO2}} = 1.005 \frac{MWh}{t_{CO2}} \tag{22}$$

Accordingly, in the developed power-to-fuel process, 1.005 MWh of thermal energy was generated per ton of $CO_2$ consumed, which can in turn be used for $CO_2$ separation. The resulting thermal coverage for the separation of $CO_2$ from the $CO_2$ sources of biogas, industrial waste gas (cement works), and ambient air are given in Table 3. For the sake of completeness, the thermal and electrical energy requirements are also listed.

**Table 3.** Thermal coverage for $CO_2$ separation from various $CO_2$ sources.

| $CO_2$ Source | Electricity Demand [MWh/t$_{CO2}$] | Heat Demand [MWh/t$_{CO2}$] | Thermal Coveragee |
|---|---|---|---|
| **Biogas (amine washing)** [66] | 0.011 | 0.631 | 159% |
| **Ambient air (direct air capture)** [65] | 0.5 | 1.5 | 67% |
| **Cement production (amine washing)** [66] | 0.2 | 1.03 | 97.6% |

The heat required to separate a ton of $CO_2$ from biogases is comparatively low, at 0.631 MWh, and so the entire thermal energy requirement can be covered by the excess medium-pressure steam and even remaining, unused medium-pressure steam. The separation of $CO_2$ from the ambient air requires the largest amount of thermal energy with 1.5 MWh per ton of $CO_2$. About 67% of this can be provided through the excess medium-pressure steam. As an example of the separation of $CO_2$ from industrial waste gases, separation from the waste gases of a cement plant by means of amine scrubbing was selected at this point. This technology requires 1.03 MWh of thermal energy to separate one ton of $CO_2$. Accordingly, 97.6% of this thermal energy can be provided through waste heat from the developed power-to-fuel process. The degrees of coverage listed in Table 3 make it clear that the developed PtF process can be coupled with $CO_2$ separation in order to provide the thermal energy required for $CO_2$ separation. In order to examine the influence of such a coupling in greater detail, Figure 14 shows the efficiencies of the entire process chain for the case of the coupling of $CO_2$ capture and the PtF process, and for the case that both processes are operated separately.

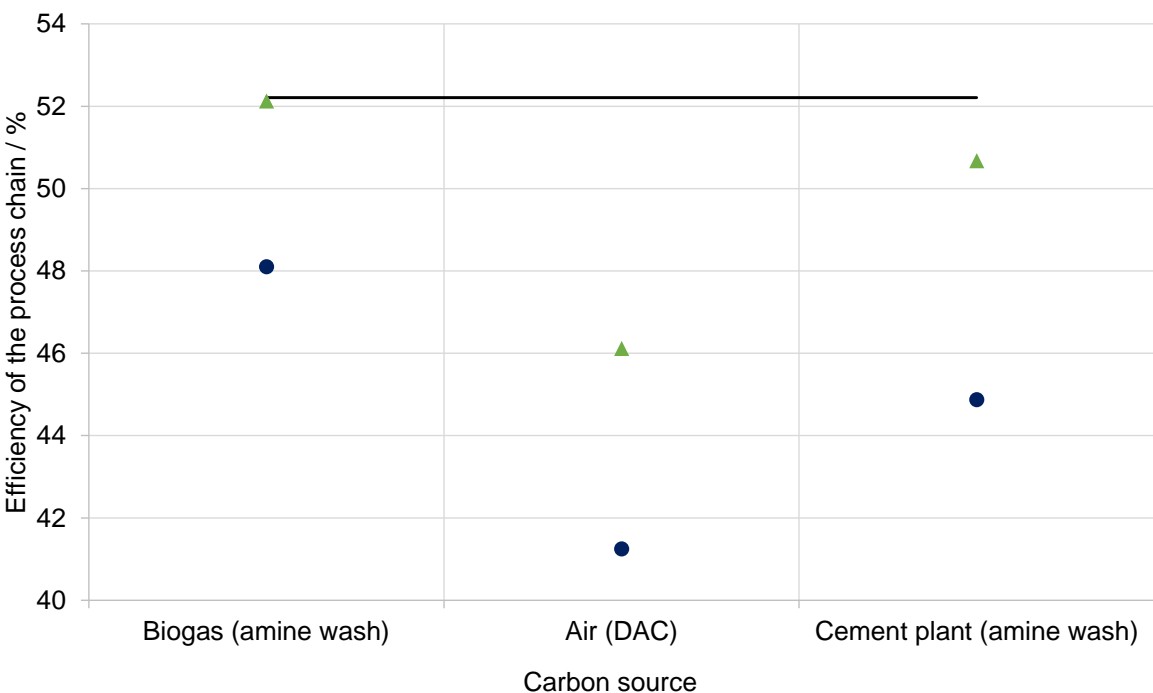

**Figure 14.** Coupling of the modeled synthesis with $CO_2$ capture technologies.

If the separation of $CO_2$ from biogas is conducted independently of the modeled PtF process, the efficiency of the process chain is approximately 48.1%. If both technologies are coupled, the entire thermal energy requirement can be provided by the fuel synthesis, so that the overall efficiency is 52.1% (i.e., almost the efficiency of the PtF process without taking $CO_2$ separation into account). The remaining difference in results from the electrical energy requirement for $CO_2$ separation are listed in Table 3. The efficiency of the entire process chain with $CO_2$ separation from ambient air was lower due to the high thermal and electrical expenditure of the technology (see Figure 14).

Nevertheless, by coupling the technologies, the efficiency can be increased from 41.3% to 46.1% (i.e., 4.8 percentage points). The greatest increase in efficiency can be seen in the coupling of the developed fuel synthesis with $CO_2$ separation from the exhaust gases of a cement works. By coupling the technologies, the efficiency can be increased by 5.8 percentage points, from 44.9% to 50.7%. Overall, it can be said that a coupling of the developed fuel synthesis with $CO_2$ separation technologies is both possible and advantageous. The excess heat incurred by the Fischer–Tropsch synthesis can be discharged from the process in the form of medium-pressure steam and used to operate various $CO_2$ separation technologies. This makes it possible to increase the efficiency of the entire process chain, from $CO_2$ to synthetic fuel, by up to around 4.87 percentage points, depending on the $CO_2$ source used.

*5.4. Comparison to a Related Analysis for E-Fuels Based on Low-Temperature Electrolysis*

In order to be able to better assess the calculated power-to-fuel efficiency of the developed fuel synthesis, it was compared with the efficiencies of alternative PtF or PtL processes. Three PtL processes developed and examined by Schemme [54] were considered. The comparison is possible, as Schemme [54] assumed the same boundary conditions and efficiencies for ancillary units as in this work. In two of the processes considered, alternative fuels were the target product, with methanol being produced in the first process and dimethyl ether (DME) in the second. In the third process considered, synthetic kerosene

and synthetic diesel were also produced through a Fischer–Tropsch synthesis, with the synthesis gas for the Fischer–Tropsch reaction being generated via a reverse water–gas shift reactor. The hydrogen supply for all three processes was provided through PEM electrolysis, for which an electrolysis efficiency of 70% was assumed. The PtL efficiencies of the three processes are compared in Figure 15 with the PtL efficiencies of the fuel synthesis developed herein for different electrolysis efficiencies. It should be noted that efficiencies are only really comparable if all of the reference points for determining them are known [67].

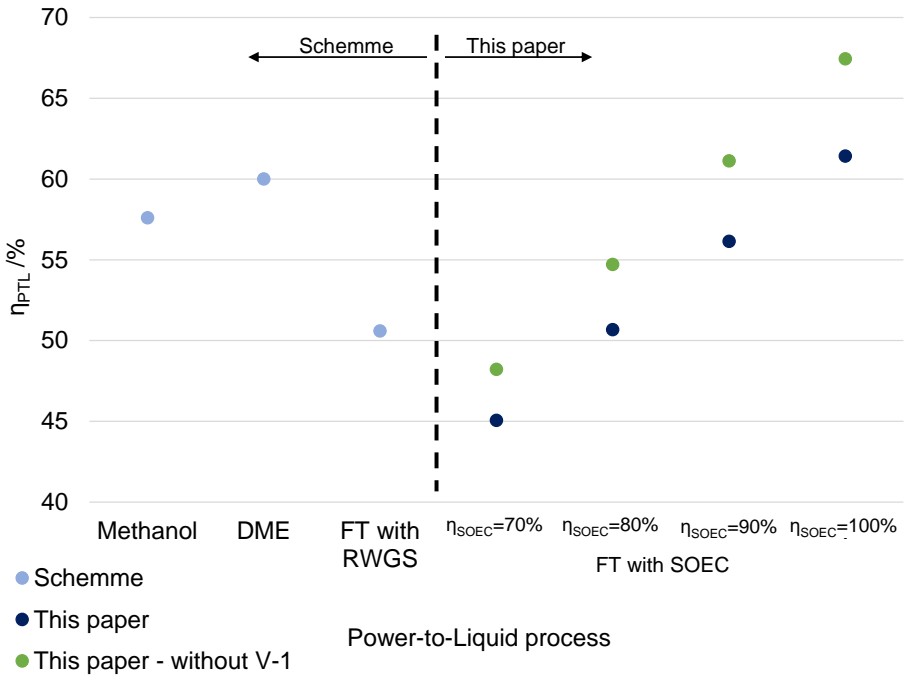

**Figure 15.** Comparison of the PtL efficiency of the developed fuel synthesis with those of alternative PtL processes ($\eta_{PTl}$ of methanol, DME, and FT with the RWGS process according to Schemme [54]).

Therefore, the following comparison should not serve as a precise quantitative assessment of the processes, but represents a qualitative classification of the developed fuel synthesis. The efficiencies shown in Figure 15 are the PtL efficiencies for the case that the processes are coupled with $CO_2$ separation from industrial exhaust gases. With the three efficiencies according to Schemme [54], $CO_2$ separation from industrial exhaust gases was also assumed, but with the difference that only an electrical energy requirement of 0.333 MWh per ton of $CO_2$ was taken into account, and the thermal energy requirement was not. Accordingly, it can be assumed that the actual PtL efficiencies of the processes from Schemme [54], taking into account the thermal energy requirement, are below the specified values. The efficiencies of the methanol and DME processes in particular were probably overestimated, as significantly less excess heat is produced in these processes than in those that employ a Fischer–Tropsch synthesis (see Schemme [54]). The PtF efficiency of the developed fuel synthesis is shown in Figure 15 for two different cases. For the standard configuration shown in blue, the PtL efficiency of around 45% for an $\eta_{SOEC}$ of 70% was below the efficiencies of all reference processes. With an electrolysis efficiency of 80%, the PtL efficiency of the developed fuel synthesis corresponded to that of the Fischer–Tropsch process with a RWGS reactor at around 51.1%. With increasing electrolysis efficiency, the PtL efficiency also further increased. With an $\eta_{SOEC}$ of 100%, the PtL efficiency of the developed fuel synthesis was approximately 61.4% and above the power-to-liquid efficiencies of the methanol and DME syntheses of 57.6% and 60%, respectively. The conclusion for this is that co-electrolysis only achieves the same PtL efficiency with a 10 percentage point higher efficiency than the process configuration of PEM electrolysis, RWGS reactor,

and Fischer–Tropsch synthesis, as assumed by Schemme [54]. However, the specified efficiencies for high-temperature electrolysis described by Peters et al. [64] partially take into account a compression of the electrolysis products to 70 bar for storage, which was not carried out in the developed fuel synthesis. Accordingly, it can be assumed that the required electrolysis power was overestimated and therefore the PtF efficiency was underestimated. To compensate for this effect, the PtF efficiency was recalculated, neglecting the compression of the synthesis gas to 30 bar by V-1 (cf. Figure 2). The resulting PtF efficiencies are highlighted in green in Figure 15. With an efficiency of the high-temperature electrolysis of 70%, the $\eta_{PTL}$ was around 48.2% at a similar level to the Fischer–Tropsch process investigated by Schemme [54], with a PEM electrolysis efficiency of 70%. The PtF efficiency of the fuel synthesis developed in this work, neglecting V-1, further increased with $\eta_{SOEC}$, with a PtF efficiency of approximately 67.4% being calculated for an $\eta_{SOEC}$ of 100%. Figure 15 thus shows the great potential of PtF processes based on co-electrolysis, and thus also the potential of the developed fuel synthesis. With an electrolysis efficiency of 70%, the high-temperature electrolysis-based process achieved similar efficiencies as the PEM electrolysis-based process. However, with high-temperature electrolysis, there is the possibility of substituting electrical energy with thermal energy and thus achieving very high electrolysis efficiencies and therefore also very high PtF efficiencies.

## 6. Techno-Economic Analysis

This section is dedicated to the analysis of the economic aspects of the developed PtF process. As the developed model is suitable for calculation with any feed stream, it is necessary to determine an order of magnitude for the process for investigation. According to a report by the Intergovernmental Panel on Climate Change [68], the cement industry emits around 932 megatons of $CO_2$ annually. For the number of cement works of 1175 given in the same source, assuming 8000 annual operating hours, this approximately corresponds to an average $CO_2$ production of 99.15 tons per hour and plant. Based on this value, a $CO_2$ feed stream of 100 tons per hour are defined for the techno-economic analysis in this work. This corresponds to a total production capacity of around 39,400 $l_{DE}$ per hour of synthetic kerosene and diesel, with a total energy consumption of around 750 MW. In the following, the methodology described by Schemme et al. and Peters et al. [69,70] was used to calculate the costs of manufacturing, whereby the investment costs for the system as well as the material and personnel costs for operating it, were first determined. The cost of the product was then determined on the basis of the specific costs. A sensitivity analysis was carried out in order to examine the influence of various cost factors on the production costs in greater detail. Finally, the calculated production costs of the developed fuel synthesis were compared with alternative PtF or PtL processes. The cost of product to be expected for the reference year 2030 was also calculated as part of the economic analysis, with all costs being converted to 2019 equivalents. Accordingly, all cost information below relates to the year 2019, unless stated otherwise.

### 6.1. Investment Cost

The first step in calculating the cost of the product was to determine the system's investment costs. The calculation of the component costs, with the exception of the reactors and electrolysis, was conducted with the Turton method using the CAPCOST Excel tool; the methodology is explained in Schemme [54]. The investment costs for the reactors and electrolysis are calculated separately. Due to the large number of components required, the investment costs for each of the system components are not discussed individually below. A detailed breakdown of the investment costs is presented in Table 4. To calculate the component costs using the Excel tool CAPCOST, the material, operating pressure, and component-specific size parameters Z must be specified. In the cost accounting, it was assumed that all system components were made of stainless steel. The operating pressures were taken from the Aspen Plus simulation, whereby an additional safety factor of 1.5 was taken into account for all pressures. The size parameters Z were determined depending on

the component group. For the pumps, compressors, and drives, the respective nominal capacities are required as size parameters for CAPCOST. These can be taken directly from Aspen Plus, taking the efficiency into account. For the cost calculation of the distillation column and the two strippers, the diameters and heights of the respective apparatuses are required. The Aspen Plus Tray sizing function was used to determine the diameter, assuming sieve trays with a distance of approximately 0.6 m (2 feet) from one another. The heights of the columns and strippers were calculated using the number of trays and the distance between them. An additional distance between the top floor and head or the lowest floor and the sump of 1.5 m was taken into account—a so-called disengagement space. The size parameter for calculating the heat exchanger corresponded to the heat transfer area of the respective heat exchanger.

**Table 4.** Investment costs for the modeled power-to-fuel process.

| Component (-Group) | Investment Cost [Mil.-€] | Share [%] |
|---|---|---|
| Pumps | 0.146 | 0.02% |
| Compressor | 122.238 | 12.87% |
| Drives | 1.680 | 1.12% |
| Columns & stripper | 1.252 | 0.13% |
| Heat exchanger | 135.524 | 14.27% |
| Vessels | 3.593 | 0.38% |
| FT reactor | 87.185 | 9.18% |
| Hydrocracker | 70.186 | 7.38% |
| Reformer | 1.862 | 0.20% |
| High-temperature water electrolyzer | 25.809 | 2.72% |
| High-temperature co-electrolyzer | 491.420 | 51.73% |
| **Sum** | **949.896** | **100%** |

In the next step, the investment costs for the three reactors were determined using the corresponding equations published Baliban et al. [71]. The calculation method presented here requires the capacities of the Fischer–Tropsch reactor, the hydrocracker, and the reformer. These can be read from the Aspen Plus simulation and are listed in Table A4 in Appendix B. It should be noted that the capacity of the Fischer–Tropsch reactor was above the permitted $s_{max}$ value and that two Fischer-Tropsch reactors were therefore operated in parallel. The determined investment costs for the reactors are shown in Table 4.

To calculate the investment costs for high-temperature electrolysis, the electrical performance of the water and co-electrolysis processes must be determined. For this purpose, as in Section 5.2, an electrolysis efficiency was calculated and the required electrolysis output determined through the calorific value of the electrolysis products. An electrolysis efficiency of 80% was assumed for the "base case." This resulted in an output of approximately 643.4 MW for the co-electrolysis and of about 33.8 MW for the water electrolysis. Table 4 gives the forecasts for the investment costs for high-temperature electrolysis for the year 2030. As part of this cost calculation, the data from Brynolf, et al. [72] was employed, as these also expressly apply to SOECs in co-electrolysis operation. The reference case was based on Brynolf et al. [72], given a mean value of €764 (≙700 € 2015) per kilowatt assumed for the investment costs for high-temperature electrolysis. The sum of the investment costs for the system components resulted in investment costs of around €949.9 mil. for the entire system. It can clearly be seen that the investment costs for the high-temperature electrolysis, totaling around €517 million and thus with a share of over 50%, made up by far the largest share of the total investment costs. Next up were the investment costs for the heat exchangers and compressors at around 14% and 13%, respectively. Costs for the Fischer–Tropsch reactor and hydrocracker made up a considerable part of the total investment costs at around 9% and 7%, respectively, whereas the costs for the remaining system components only played a subordinate role, at less than 2%. Due to the high share of investment costs for high-temperature electrolysis, the influence of the electrolysis efficiency and investment

costs for the electrolysis against the cost of the product is discussed in greater depth in Section 6.4.3.

### 6.2. Material and Personnel Costs

The material costs were derived from the costs of the raw materials required for production and those for the required resources. The amounts of the respective materials can either be taken directly from the Aspen simulation or determined using the data shown in Figure 11. If the required quantities for raw materials and operating resources are known, the annual material costs are determined using a specific price for the respective material. An annual operating time of 8000 h was assumed for calculation of the costs. The calculated material costs for the raw materials are presented in Table 5. For the $CO_2$ price, the base case was assumed to be €70 per ton. This is the lowest price that can be expected in the short- to medium-term for $CO_2$ that is separated from the exhaust gases of a cement plant (see Schemme [54]). Overall, it can be seen that the costs for $CO_2$ make up the largest part of the raw material costs, at €56 million out of a total of around €64 million. The influence of the costs of $CO_2$ on the production costs was considered in the sensitivity analysis.

**Table 5.** Raw material costs for the PtF process.

| Raw Material | Demand | Spec. Costs | Material Cost |
| --- | --- | --- | --- |
| Carbon dioxide | 100.0 t/h | 70.0 €/t [1] | 56,000,000 €/a |
| Oxygen | 13.4 t/h | 70.0 €/t [2] | 7,491,344 €/a |
| Process water | 157.2 t/h | 0.1 €/t [3] | 125,753 €/a |
| $C_R$ | - | - | 63,617,097 €/a |

[1] Brynolf et al. [72]. [2] Cheaper also in the literature: Fraunhofer-Institut [73]. 70 €/t was chosen to not underestimate the costs. [3] Cheaper also in the literature: INTRATEC [74]. 0.1 €/t was chosen to not underestimate the costs.

The annual material costs for the required equipment are presented in Table 6. For the base case of the cost calculation, an electricity price of €40 per MWh was assumed, which was the approximate average electricity exchange price in Germany for 2019 according to the 'energy charts' provided by the Fraunhofer Institute [75]. Table 6 shows that electricity costs made up the majority of operating costs. Therefore, Section 6.4.1 examines in greater detail how the price of electricity influences the cost of the product.

**Table 6.** Operating costs for the power-to-fuel process.

| Raw Material | Consumption | Specific Costs | Material Costs |
| --- | --- | --- | --- |
| Cooling water | 3324.6 t/h | 0.1 €/t [1] | 2,659,653 €/a |
| Cooling air | 42,800.8 t/h | - | - |
| Electricity | 752.65 MW | 40 €/MWh [2] | 240,848,000 €/a |
| $C_B$ | - | - | 243,507,653 €/a |

[1] Cheaper also in the literature: INTRATEC [74]. 0.1 €/t was chosen not to underestimate costs. [2] 'Energy charts' [75].

To calculate the annual personnel costs $C_p$ using Equation (23), the number of work steps with particulate solids P and the number of system components to be monitored or controlled $N_{NP}$ must be determined. In the developed power-to-fuel process, no work steps were carried out with particulate solids, and P was accordingly zero. The number of system components to be monitored and the total $N_{NP}$ were as follows: 65 compressors, one column, 181 heat exchangers, three reactors, and 14 electrolyzers.

The high number of compressors and heat exchangers compared to the modeling resulted from the fact that in the Excel tool CAPCOST, the highest possible compressor output of a single compressor was limited to 3000 kW, and the largest possible heat exchanger surface of a single heat exchanger to 1000 m$^2$. The required electrolysis power for the base case was determined to be approximately 643.4 MW for the co-electrolysis and 33.8 MW for the water electrolysis processes. According to Brynolf et al. [59], the maximum

output by an SOEC to be expected by 2030 is 50 MW. Accordingly, a total of 14 SOEC units are required for the power-to-fuel process. This results in a value of 264 for $N_{NP}$. The average annual gross salary of a full-time employee in the chemical industry in Germany in 2019 was €58,896; see [76]. Taking into account the non-wage labor costs of approximately 23% (see [77]), the annual personnel costs can be calculated using Equation (23), whereby PAYROLL is determined as €76,488 according to [76,77].

$$
\begin{aligned}
C_P &= 5.38 \cdot \sqrt{6.29 + 31.7 \cdot P^2 + 0.23 \cdot N_{NP}} \cdot PAYROLL \\
&= 5.38 \cdot \sqrt{6.29 + 31.7 \cdot 0^2 + 0.23 \cdot 264} \cdot 76,488 € \\
&\approx 3,365,561 €
\end{aligned}
\tag{23}
$$

### 6.3. Product Cost

Using Equation (24), annual production costs were calculated. However, a depreciation period t and an interest rate i must first be specified in order to determine the annuity. According to Brynolf et al. [59], lifetimes of between 10 and 20 years are to be expected for SOEC systems and maximum lifetimes of less than 90,000 operating hours for SOEC stacks. According to Schmidt et al. [78], for SOEC stacks, maximum operating times of over 100,000 h or, according to one of the experts questioned, of just 30,000 h, can be expected. A depreciation period of 12 years was assumed for the base case of the cost calculation, which corresponded to a total of approximately 96,000 operating hours with an annual operating time of 8000 h. Different values can be found for the interest rate in the literature for power-to-fuel processes such as an interest rate of 4% in Schmidt, et al. [79] and 8% in Buddenberg et al. [80]. An interest rate of 5% was selected for the base case of cost accounting. The influence of the selected depreciation period and selected interest rate was examined as part of the sensitivity analysis. If the depreciation period and interest rate are selected, the annual production costs can be calculated according to Equation (24):

$$
\begin{aligned}
COM &= 0.151 \cdot FCI + 2.284 \cdot C_P + 1.031 \cdot (C_R + C_B) \\
&\quad + FCI \cdot \left( \frac{i \cdot (1+i)^t}{(1+i)^t - 1} + i \cdot 0.15 \right) \\
&= 582.1 \ Mil.€
\end{aligned}
\tag{24}
$$

With:

Investment costs $FCI$ = €949.9 million, see Table 4;
Personnel costs $C_p$ = €3.37 million, see Equation (23);
Raw material costs $C_R$ = €63.6 million, see Table 6;
Operating costs $C_B$ = €243.5 million, see Table 6;
Depreciation period $t$ = 12;
Interest rate $i$ = 0.05.

For the specified feed stream of 100 tons of $CO_2$ per hour, the production output of the power-to-fuel process totaled around 39,400 L of diesel equivalent per hour. The specific production costs result from Equation (25):

$$
com = \frac{582.1 \cdot 10^6 \frac{€}{a}}{39400 \frac{l_{DE}}{h} \cdot 8000 \frac{h}{a}} \approx 1.85 \frac{€}{l_{DE}}
\tag{25}
$$

Table 7 shows the cost allocation based on Otto [81]. As a result, the base case under consideration resulted in fuel production costs of around €1.85 per liter of diesel equivalent. Several important influencing factors on the cost of product could also be identified. At almost 42%, the operating resources made up the largest share of fuel production costs. It can be seen in Table 6 that the majority of operating costs were caused by electricity costs, and therefore the electricity price exerts a strong influence on the cost of the product.

**Table 7.** Distribution of the fuel production costs according to Otto [81].

| Cost Component | Specific Production Cost [€/$l_{DE}$] | Share [%] |
|---|---|---|
| Raw materials | 0.20 | 10.9 |
| Resources | 0.77 | 41.9 |
| Overhead: transportation, storage, etc. | 0.12 | 6.3 |
| Manufacturing personnel | 0.01 | 0.6 |
| Surveillance and office staff | <0.01 | 0.1 |
| Maintenance and repair work | 0.19 | 9.8 |
| Auxiliary materials | 0.03 | 1.5 |
| Laboratory costs | <0.01 | 0.1 |
| Patent and license fees | 0.04 | 2.4 |
| Taxes and insurance | 0.10 | 5.2 |
| Administrative costs | 0.03 | 1.6 |
| Annuity | 0.36 | 19.6 |
| **Total** | 1.85 | 100 |

The annuity accounted for the second largest share of fuel production costs, the amount of which depends on the selected interest rate and the selected depreciation period as well as on the investment costs FCI. In addition, the FCI have, according to the cost allocation by Otto [81], a direct impact on several other cost components such as maintenance and repair work. Over 50% of the total investment costs are made up by those for high-temperature electrolysis. Accordingly, it can be assumed that the investment costs for the electrolysis have a major influence on fuel production costs. The investment costs for high-temperature electrolysis depend, on one hand, on the specific investment costs per kilowatt of power and, on the other, on the efficiency of the high-temperature electrolysis. The last important cost factor can be identified as the $CO_2$ costs, as these account for almost 90% of the raw material costs (see Table 2) and so almost 10% of the total cost of the product. Overall, the electricity price, efficiency of high-temperature electrolysis, specific investment costs for high-temperature electrolysis, depreciation period, and interest rate as well as the price of $CO_2$ are identified as important cost items. The influence of these factors is examined in more detail in the following section.

*6.4. Sensitivity Analysis*

A sensitivity analysis is presented in this section to examine the influence of the cost factors identified in Sections 6.1–6.3. For this purpose, the assumptions for the electricity price, electrolysis efficiency, specific investment costs for the electrolysis, depreciation period, and interest rate as well as the $CO_2$ costs were varied and the fuel production costs calculated. The other cost factors were left at the values assumed for the base case (see Table 8). Finally, the influences of the respective cost factors were compared in the form of a so-called tornado diagram.

**Table 8.** Assumptions for the base case.

| Cost Parameter | Assumption |
|---|---|
| Electricity price | 40 €/MWh |
| $\eta_{SOEC}$ | 80% |
| Spec. investment electrolysis | 764 €/kW |
| Depreciation period | 12 years |
| Interest rate | 5% |
| $CO_2$ cost | 70 €/t |

6.4.1. Influence of the Electricity Price

Figure 16 displays the production costs in relation to the assumed electricity price. For the sake of clarity, the cost components "production staff", "monitoring and office staff",

"auxiliary materials", "laboratory costs", and "patent and license fees" were combined into "other production costs" in this and the following figures.

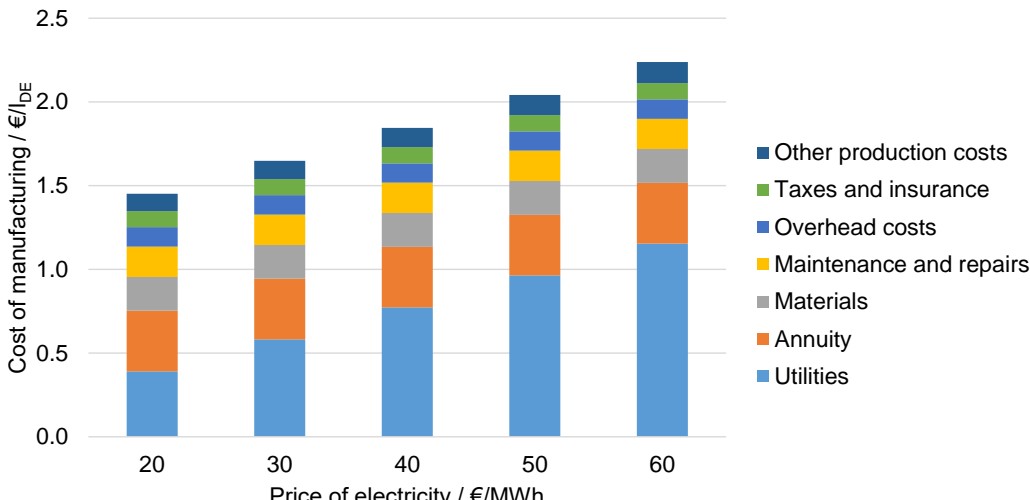

**Figure 16.** Production costs depending on the electricity price.

The electricity price was varied based on the €40 per megawatt hour assumed for the base case plus or minus 50%. For €20 per megawatt hour, the cost of the product was approximately €1.5/$l_{DE}$, and for €60 per megawatt hour, it was around €2.2/$l_{DE}$. The operating costs, which mainly consisted of electricity, made up around 25% at €20/MWh and over 50% of the cost of the product at €60/MWh. The share of electricity costs in the production costs and, accordingly, their influence on the production costs, was very high. Thus, the local electricity price should be considered as an important criterion when choosing locations.

6.4.2. Influence of Electrolysis Efficiency

Figure 17 shows the fuel production costs as a function of the electrolysis efficiency. In Section 6.2, it was noted that electrolysis efficiencies of less than 70% are not to be expected for the developed fuel synthesis. Therefore, the efficiency of high-temperature electrolysis was varied from 70% to 100% for the sensitivity analysis. For an efficiency $\eta_{SOEC}$ of 70%, the cost of product was around €2.0/$l_{DE}$. The levelized costs of the product decreased with increasing electrolysis efficiency, and for an efficiency $\eta_{SOEC}$ of 100%, the levelized costs of the product were around €1.6/$l_{DE}$. It can be seen in Figure 17 that with increasing electrolysis efficiency, both the operating material costs and cost items that are dependent on the investment costs for the system fell. This was due to the fact that the required electrolysis power decreased with increasing electrolysis efficiency, which in turn reduced the investment costs required for the electrolysis and its power consumption. As stated above, the electrolysis efficiency is strongly influenced by the extent to which high-temperature heat is available for the operation of the water and co-electrolysis. Accordingly, when choosing a location for fuel synthesis, it should be determined whether such high-temperature heat is generated in an existing system, and if heat coupling is possible.

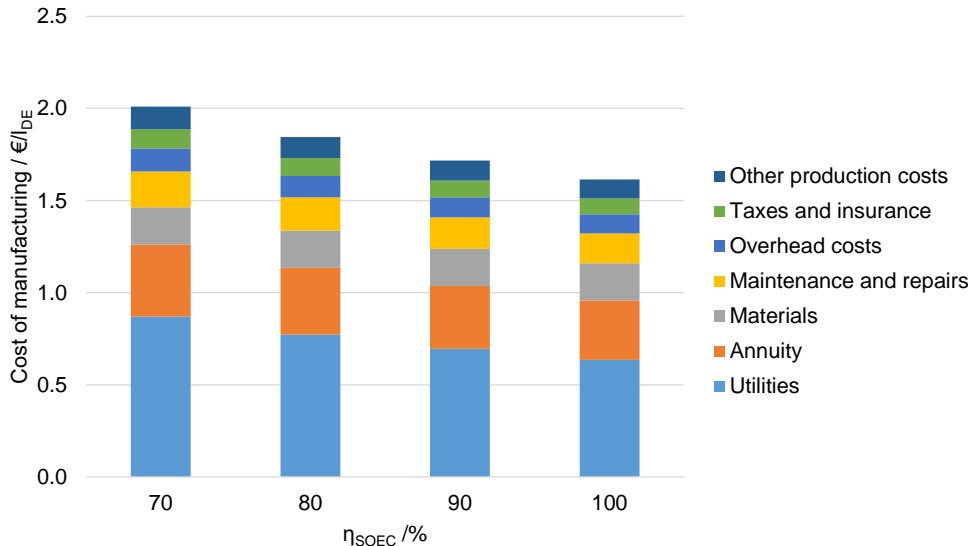

**Figure 17.** Product production costs depending on the electrolysis efficiency $\eta_{SOEC}$.

6.4.3. Influence of the Specific Investment Costs for Electrolysis

Figure 18 shows the cost of the product as a function of the specific investment costs for water and co-electrolysis. Brynolf et al. [72] provide a value of €436/kW ($\triangleq$400 €/kW @ 2015) as the lower limit and a value of €1091/kW ($\triangleq$1000 €/kW @ 2015) as the upper limit for the specific investment costs for high-temperature electrolysis expected by 2030 (see Table 4). For the sensitivity analysis, the specific investment costs were varied accordingly. The cost of product for €436/kW was around €1.7/$l_{DE}$, and for €1091/kW, there was a product cost of around €2.0/$l_{DE}$. This corresponded to a deviation from the fuel production costs for the base case of €764/kW of around ±10%. The influence of the specific investment costs was, accordingly, significantly less than that of the electricity price, for instance. However, it must be noted that the projection of Brynolf et al. [72] only applies in the event that major technical advances are made by 2030 and are therefore to be considered target values. It is important to invest in research and development so that the developed fuel synthesis or other power-to-fuel processes based on SOEC technology can be implemented in the future.

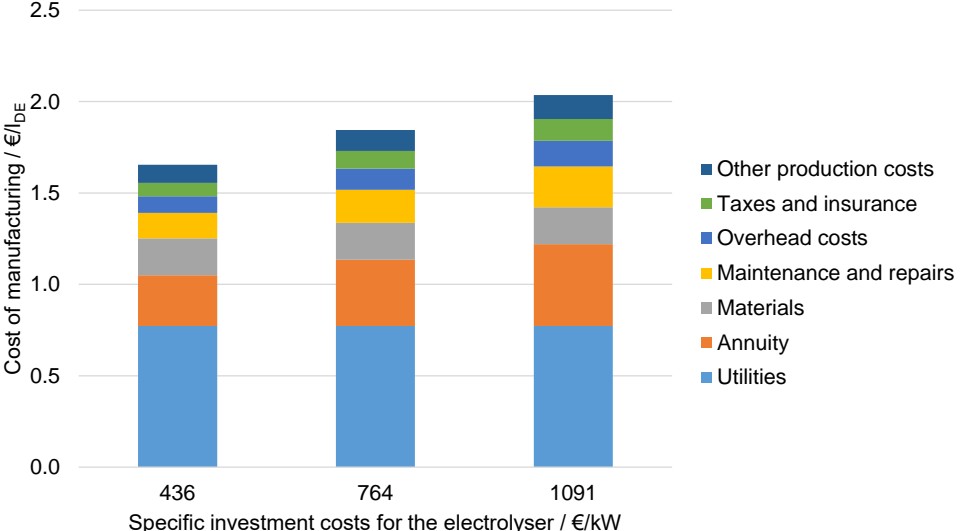

**Figure 18.** Product production costs as a function of electrolyzer investment costs.

### 6.4.4. Influence of the Depreciation Period and Interest Rate

The cost of product as a function of the selected depreciation period and selected interest rate are shown in Figure 19. The cost of product for a short depreciation period of nine years with a high interest rate of 7% and a long depreciation period of 15 years with a low interest rate of 3% were compared with the base case of 12 years and 5%, respectively. In the worst case, with a short depreciation period and high interest rate, the cost of product was €2.0/$l_{DE}$, and in the favorable case with a long depreciation period and low interest rate, the cost of product was €1.7/$l_{DE}$. This deviation from the base case resulted from the annuity, which was reduced by almost 45% from the unfavorable case to the favorable one. Accordingly, the location-dependent investment conditions should also be taken into account when choosing a location.

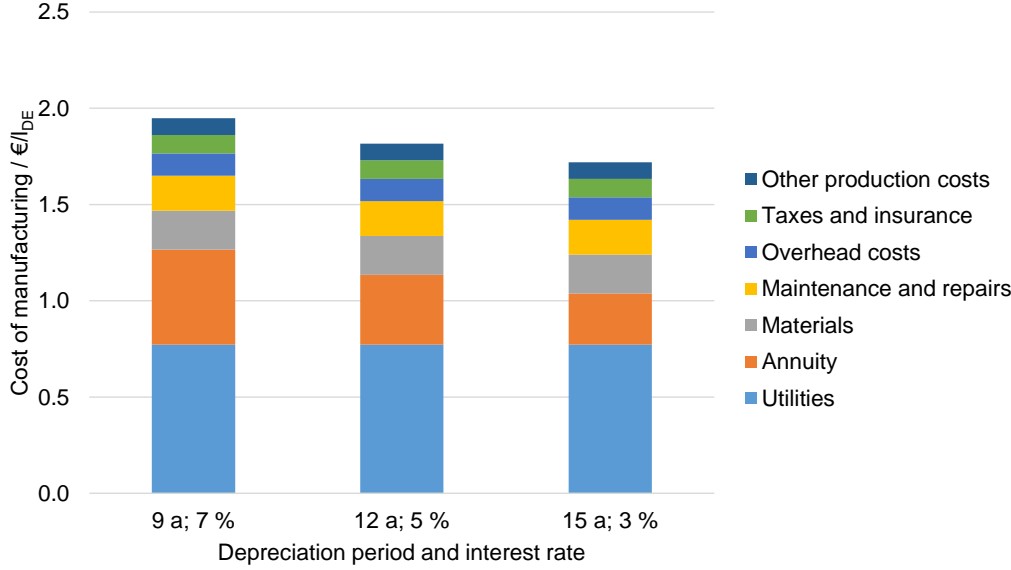

**Figure 19.** Product production costs depending on the depreciation period and interest rate.

### 6.4.5. Influence of $CO_2$ Costs

Figure 20 shows the cost of the product as a function of that of the required $CO_2$. According to Brynolf and Taljegard [37], the expected long-term costs for $CO_2$ that are separated from the exhaust gases of a cement plant are between €30 and €50 per ton. Accordingly, for the sensitivity analysis, production costs in this range were assumed. For the sake of completeness, the cost of the product was also calculated to be between €70 and €90 per ton of $CO_2$. The cost of the product was around €1.7/$l_{DE}$ and €1.9/$l_{DE}$. It is clear that the $CO_2$ price has an influence on the production costs, but that it is significantly lower compared to the other cost factors examined. This means that a favorable $CO_2$ price in the range considered is advantageous for fuel synthesis, but not absolutely essential.

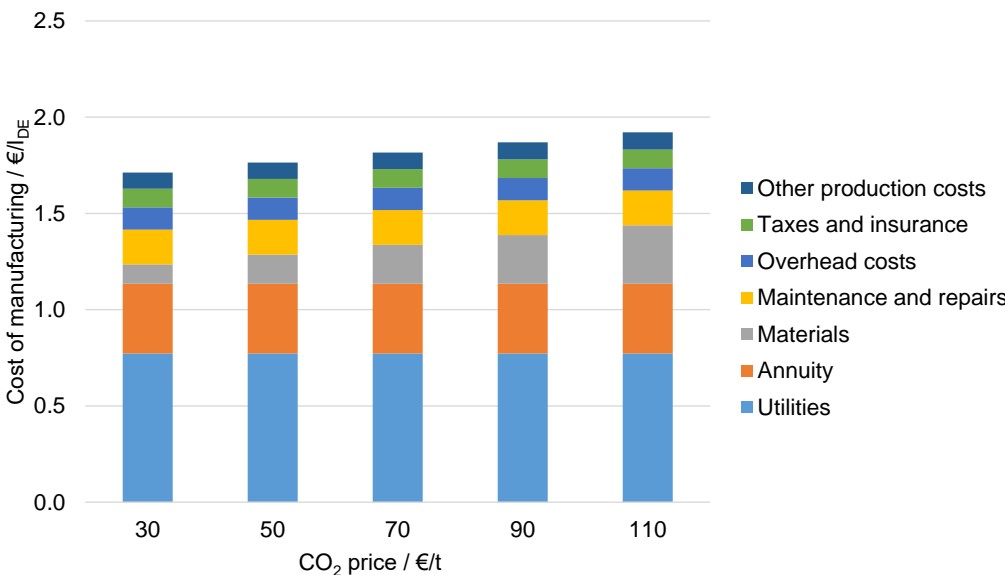

**Figure 20.** Product production costs as a function of the $CO_2$ price.

6.4.6. Influence of the Cost Factors in Comparison

A tornado diagram, which can be seen in Figure 21, was used to compare the influence of the respective cost factors on the levelized cost of electricity (LCOE). This diagram compares the areas in which the levelized product costs are dependent on the respective cost factors, based on the base case of €1.85/$l_{DE}$. For this comparison, the best case and the worst case assumptions from the previous sections were used and highlighted together with the base case assumptions in Figure 21. In Figure 21, it is once again clear that the electricity price had the greatest influence on the fuel production costs and could influence fuel production costs up or down by up to 40 ct/$l_{DE}$ for the price range examined. It is important to note that the tornado diagram only considered the cost parameters individually. In reality, it can be assumed that there will be a mixture of "best", "base", and "worst" cases presented. However, due to the sum of all possible savings when all of the best cases arrive, a lower limit for the fuel production costs was given at €0.94/$l_{DE}$. In the same manner, with the arrival of all worst cases with €2.95/$l_{DE}$, there was an upper limit for the cost of the product.

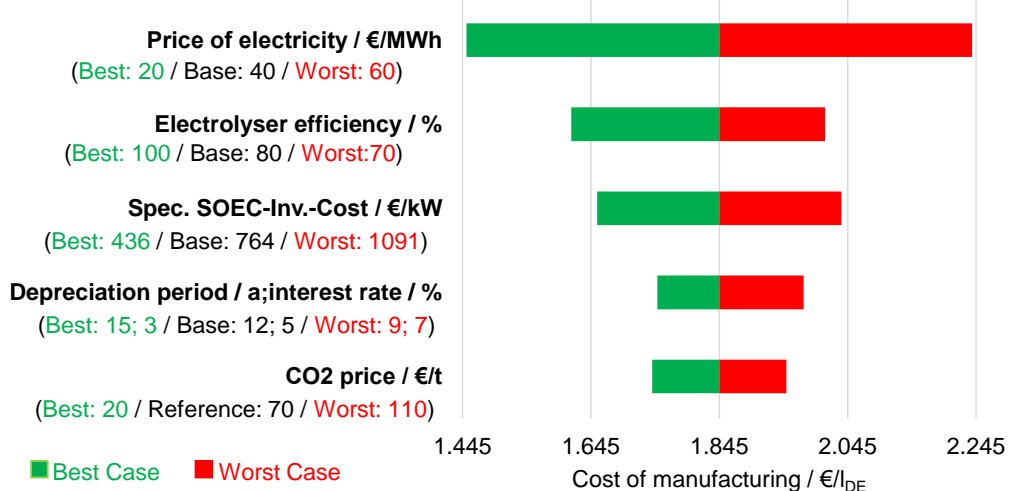

**Figure 21.** Tornado diagram of the sensitivity analysis of product generation costs.

### 6.5. Comparison with Alternative Power-to-Liquid Processes

In the following section, the calculated production costs are compared with those of alternative power-to-fuel processes. As noted in Section 6.4, the methanol and DME synthesis investigated by Schemme [54] and the Fischer–Tropsch synthesis in combination with a RWGS reactor were used for the comparison. The production costs of the three processes are compared in Figure 22 with the manufacturing costs for the worst, base, and best cases specified in Section 6.4.

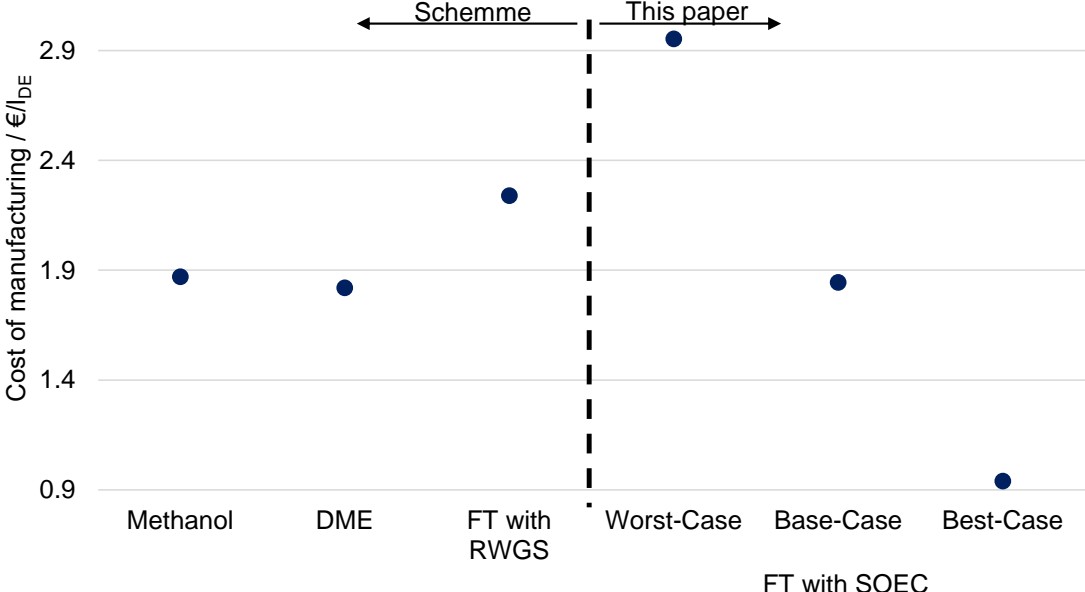

**Figure 22.** Comparison of the production costs of the developed fuel synthesis with those of alternative power-to-liquid processes (costs of methanol, DME, and FT with the RWGS process according to Schemme [54]).

It must be taken into account that with respect to Schemme [54], different framework conditions were selected in some cases than in the economic analysis carried out in this study. In the three comparison processes, for example, the required hydrogen was not produced in the process, but was obtained externally and taken into account in the cost calculation at a price of €4.6/kg of $H_2$. Therefore, the following comparison serves as a qualitative classification of the calculated manufacturing costs rather than an exact quantitative comparison of the production costs of the different power-to-liquid processes. In addition, the processes investigated by Schemme [54] were smaller than the fuel synthesis investigated herein.

For example, the Fischer–Tropsch reactor investigated by Schemme [54] was fed with a feed stream of approximately 245,000 $Nm^3$/h, whereas in the economic analysis carried out in this study, approximately 350,000 $Nm^3$/h flowed into the Fischer–Tropsch reactor. It is therefore possible that there is a potential for savings through scaling effects for the processes examined by Schemme [54]. However, Figure 22 clearly shows the great potential of the fuel synthesis that has been developed, also taking into account these potential savings. The production costs of €1.85/$l_{DE}$ are already competitive for the base case with the fuel production costs of the methanol and DME synthesis of €1.87/$l_{DE}$ and €1.82/$l_{DE}$, respectively. In the best case, the costs can even be undercut with €0.94/$l_{DE}$. An additional advantage of the developed fuel synthesis compared to methanol and DME synthesis is that the infrastructure for traditional fuels already exists and further costs can therefore be saved.

## 7. Conclusions

Power-to-fuel technology represents a promising possibility for making the transport sector $CO_2$-neutral in the future. An especially interesting power-to-fuel concept is the coupling of high-temperature co-electrolysis with Fischer–Tropsch synthesis, as this carries some thermodynamic advantages. The aim of this study was to develop such a power-to-fuel process, model the developed process in a process simulation program, and then carry out a techno-economic analysis of the overall process. In the developed fuel synthesis, the entire process chain was considered, starting with water and $CO_2$ and ending with the fuel according to specifications. First, water and $CO_2$ were converted into synthesis gas, consisting of hydrogen and carbon monoxide, by means of high-temperature co-electrolysis. In the next step, the synthesis gas was converted into hydrocarbons through a Fischer–Tropsch synthesis, and then processed into synthetic diesel according to EN 15940 and synthetic kerosene type FT-SPK according to ASTM 7566. The fuel preparation consisted of a hydrocracker, reformer, and carrier steam distillation. An additional high-temperature water electrolysis system was used to provide the hydrogen required for the hydrocracker. The process simulation was implemented in the simulation program Aspen Plus, whereby the model was designed for the calculation of any mass flows and so any system sizes. In addition, an energy integration analysis was conducted. The results of the process simulation provide information regarding the material and energetic balance of the process. In the developed fuel synthesis, 1 L of diesel equivalent (35.9 MJ) of synthetic fuels was produced, which was then broken down energetically into 38.9% kerosene and 61.1% diesel. An examination of the fuels produced indicated that both synthetic diesel and synthetic kerosene meet the requirements of the above standards. To produce one liter of diesel equivalent, 2.54 kg of $CO_2$, 3.99 kg of water, and 0.34 kg of oxygen are required. The energetic analysis of the process shows that the energy requirement of the high-temperature co-electrolysis was reduced by the energy integration from about 75 MJ/$l_{DE}$ over 20% to about 59 MJ/$l_{DE}$. This makes it clear that the coupling of Fischer–Tropsch synthesis with high-temperature electrolysis represents an attractive power-to-fuel concept. In addition, it was found that the energy requirement of the process and so the power-to-liquid efficiency depends heavily on the efficiency of the electrolysis. The power-to-liquid efficiency for an electrolysis efficiency of 70% was approximately 46%, and with an electrolysis efficiency of 100%, the PtL efficiency was almost 67%. The assumed base case electrolysis efficiency of 80% resulted in a PtL efficiency of 52%, whereby the electrical energy for the co-electrolysis, with about 59 MJ/$l_{DE}$, made up more than 85% of the total energy requirement of about 69 MJ/$l_{DE}$. Accordingly, the co-electrolysis represents the critical element of the developed fuel synthesis and presents itself as a topic for further research in order to develop a better understanding of the technology as well as to identify possible energy-saving potentials. The energetic analysis also showed that the developed power-to-fuel process generated an excess heat of around 1.005 MWh per ton of $CO_2$ consumed. This heat can be used for $CO_2$ capture technologies. This study showed that the excess $CO_2$ could cover around 67% of the thermal energy required to separate a corresponding amount of $CO_2$ from the ambient air and around 97% of the thermal energy requirement for separating $CO_2$ from industrial waste gases (cement works). The thermal energy requirement of $CO_2$ separation from biogas can be fully covered. This option is very attractive since it offers a biogenic $CO_2$ source, resulting in a completely sustainable route. In the long-term, the further technical development of the direct separation from ambient air will surely enable a broad application possibility for the developed technology. The thermal coupling of the power-to-fuel process with $CO_2$ capture technologies therefore represents a good opportunity to improve the efficiency of the entire process chain, from $CO_2$ to synthetic fuel. If fuel synthesis is coupled with $CO_2$ separation from biogases, the overall efficiency can be increased from 48.1% to 52.1% (i.e., by four percentage points). In the case of $CO_2$ separation from the ambient air, the thermal coupling can increase the efficiency by 4.8 percentage points, from 41.3% to 46.1%. The largest increase in efficiency was found when the fuel synthesis was coupled with the separation of $CO_2$ from industrial exhaust

gases. These values show the importance of the $CO_2$ capture technology and its relevance for the overall process. In order to maximize the efficiency of the future demonstration projects, it is therefore important to select the optimal site concerning the $CO_2$ potential. The process model developed in this study can be used to analyze the economic viability of the examined locations given that, as already described, it is suitable for calculating any system parameters. The further development of the outlined process model can also be the subject of future research. For example, the models developed for calculating the co-electrolysis or Fischer–Tropsch reactor could be enhanced by kinetic models.

In order to realize the potential of the developed fuel synthesis, two important considerations are necessary. On one hand, it is important to invest in the research and development of SOEC technology and to increase the low TRL of the SOEC and bring high-temperature electrolysis to a megawatt scale and market maturity. On the other hand, the choice of location plays a decisive role. In this way, the great potential of the developed fuel synthesis can, above all, be realized in locations where cheap electricity and high-temperature heat are available for the operation of the SOEC.

**Author Contributions:** Conceptualization: R.P., R.C.S. and N.W.; Methodology: R.P., F.S. and R.C.S.; Literature review and writing: N.W.; Process analysis ASPEN PLUS: N.W. and F.S.; Techno-economic analyses: F.S., N.W. and D.S.; Technical writing—original draft preparation: N.W.; Writing and visualization—review and editing: All authors; Supervision: R.P., J.R., M.G. and D.S. All authors have read and agreed to the published version of the manuscript.

**Funding:** This study received no external funding.

**Acknowledgments:** The authors would like to thank all members of the Institute for Electrochemical Process Engineering (IEK-14) for fruitful discussions on all of the technologies of the energy transition and their role in the future energy system. Special thanks to the research groups of Ro. Peters and Q. Fang on SOCs, systems engineering, and electrochemistry for valuable discussions related to technical data on SOECs. The exchange of information with D. Schäfer deserves special mention. Additionally, we thank C. Wood for the language editing.

**Conflicts of Interest:** The authors have no conflict of interest to disclose.

## Abbreviations

| | |
|---|---|
| SRK | Equation of State (EOS, cubic) Soave–Redlich–Kwong |
| RKS-BM | Soave–Redlich–Kwong EOS with Boston–Mathias Alpha-Function |
| BK10 | Braun K-10 |
| NRTL-RK | Non-Random-Two-Liquid, extended by EOS Redlich–Kwong for calculation of the gas phase |

## Appendix A. Effect of Chain Growth Probability on the Product Distribution

In order to illustrate the influence of the chain growth probability on the product distribution, Figure A1 compares the product distribution of a Fischer–Tropsch synthesis for $\alpha = 0.88$ and $\alpha = 0.92$ for $C_1$ to $C_{60}$ according to the Anderson–Schulz–Flory distribution (Equation (12)). The course of the mass fractions of the various hydrocarbons was similar in both curves. First, the mass fractions increased with increasing chain length until a maximum was reached and the curve fell again. However, the curve for $\alpha = 0.88$ rose much more rapidly in the area of short hydrocarbon chains and already reached the maximum at a chain length of $n = 8$. In addition, it flattened out very steeply, which means that the proportion of long-chain hydrocarbons in the product was very low. In comparison, the curve for $\alpha = 0.92$ rose much more slowly and only reached the maximum at a chain length of $n = 12$, and then flattened out more slowly. Accordingly, the proportion of long-chain hydrocarbons in the product of the Fischer–Tropsch synthesis was significantly higher for $\alpha = 0.92$, or for high chain growth probabilities in general. It follows that to maximize kerosene and diesel production in a power-to-fuel process, the chain growth probability of the Fischer–Tropsch synthesis should also be maximized.

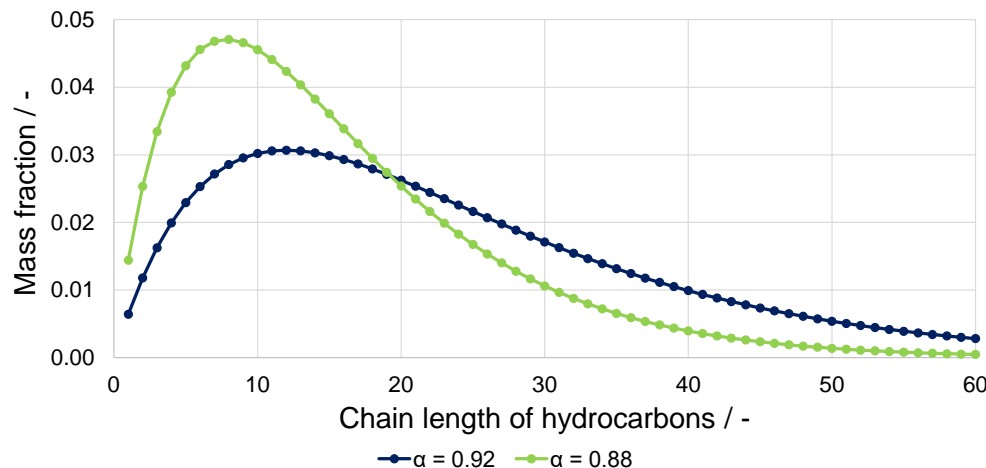

**Figure A1.** Product distribution of the Fischer–Tropsch synthesis for α = 0.88 and α = 0.92.

## Appendix B. Key Data from Process Simulations

**Table A1.** Energy-specific material balance of the developed fuel synthesis.

| Input | | Output | |
|---|---|---|---|
| **Carbon dioxide** | 2.54 kg | Synthetic fuel | 1 $l_{DE}$ |
| **Water** | 3.99 kg | Share kerosene | 38.9% |
| **Oxygen** | 0.34 kg | Share diesel | 61.9% |
| | | Water | 2.89 kg |

**Table A2.** Resources used.

| Resources | Temperature Range | Specific Heating-/ Cooling-Power |
|---|---|---|
| **Low-pressure saturated steam** | 124–125 °C | 2193 kJ/kg |
| **Medium-pressure saturated steam** | 174–175 °C | 2036 kJ/kg |
| **Cooling water** | 20–25 °C | 21 kJ/kg |
| **Cooling air** | 30–35 °C | 5 kJ/kg |
| **Electricity** | - | - |

**Table A3.** Energy-specific resource balance of the developed fuel synthesis. Negative values in the sum mean that excess steam is available from the process. See Table A2 for details on the resources used.

| Resources | Quantity [kg/$l_{DE}$] | Energy [MJ/$l_{DE}$] |
|---|---|---|
| **Low-pressure saturated steam** | | |
| **Produced** | −0.235 | −0.515 |
| **Demand** | 0.196 | 0.429 |
| **Sum** | −0.039 | −0.086 |
| **Medium-pressure saturated steam** | | |
| **Produced** | −6.059 | −12.339 |
| **Demand** | 1.546 | 3.148 |
| **Sum** | −4.513 | −9.191 |
| **Cooling water** | | |
| **Demand** | 84.370 | 1.764 |
| **Cooling air** | | |
| **Demand** | 1086.192 | 5.431 |
| **Electricity (w/o electrolysis)** | | |
| **Demand** | - | 6.890 |

**Table A4.** Capacities of the Fischer–Tropsch reactor, the hydrocracker and the reformer.

|  | FT Reactor | Hydrocracker | Reformer |
|---|---|---|---|
| Educt-/product rate | 350,322 $Nm^3$/h | 1191 t/d | 159,483 $Nm^3$/h |
| $s_{max}$-value | 228,029 $Nm^3$/h | 6256 t/d | 9,438,667 $Nm^3$/h |
| Number of reactors | 2 | 1 | 1 |

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
