# Peer review of "A Techno-Economic Assessment of Fischer–Tropsch Fuels Based on Syngas from Co-Electrolysis"

_processes, doi:10.3390/pr10040699_

Round 1

Reviewer 1 Report

Interesting paper for techno-economic assessment.  I have the following recommendations for the work:

  • In point 7 – „ Techno-economic analysis „ - it is a little difficult to separate what is based on other research and what is the author's contribution .

To expand the conclusion. There are interesting conclusions in the presentations of the paper, which can be given briefly in paper.

In conclusion, Interesting research and I propose that it be published.

Author Response

Dear reviewer,
thank you for your valuable comments.

  • In point 7 – „ Techno-economic analysis „ - it is a little difficult to separate what is based on other research and what is the author's contribution .

We have rephrased this part and added a further source to clearly present which assumptions come from other sources, where the methodology comes from and highlighted that the main information base for the economic analysis comes from the technical analysis, namely Aspen Plus simulation results, which is an important part of this study.

  • To expand the conclusion. There are interesting conclusions in the presentations of the paper, which can be given briefly in paper.

The conclusions were extended by points addressing the role and the future prospects of the CO2 separation technology. Furthermore, some aspects from the comparison part were shifted to the conclusions.

Reviewer 2 Report

In this paper, a "A techno-economic assessment of Fischer-Tropsch fuels based on syngas from co- and CO2-electrolysis" is proposed. This study focuses on the coupling of Fischer-Tropsch synthesis for the production of synthetic diesel and kerosene to a high-temperature electrolysis unit. As a part of worldwide efforts to substantially reduce CO2 emissions, power-to-fuel technologies offer a promising path to make the transport sector CO2-free, complementing the electrification of vehicles. The author's research needs to be improved, and the following suggestions are put forward:

  1. Please check if "co- and CO2-electrolysis" is correct in the title.
  2. The background part of section II lacks insights into the process of power conversion into fuel.
  3. The method proposed in 3.6 for technical and economic evaluation has been relatively mature and has no novelty.
  4. Section 4 gives an overview of the current situation of fluid power system. It is suggested to increase the length and add some schematic diagrams to illustrate the demonstration process for readers' understanding.
  5. Section 4 does not fully reflect the "verification of the technical maturity of each component in the developed fuel synthesis", which should be supplemented.
  6. Section 6 places more emphasis on the discussion part. If possible, please introduce the simulation in detail and compare the system response to support this proposition.
  7. In Section 8, only one point is put forward. For the theme of future research, it is suggested to make more prospects for the fields to be further studied.
  8. Further elaboration of the images in Appendix A is recommended.

Author Response

Dear reviewer,
thank you for your valuable comments. Please have a look on our changes that we made to improve the paper.

  1. Please check if "co- and CO2-electrolysis" is correct in the title.

We deleted CO2- electrolysis from the title, since co-electrolysis is the core technology used in our approach.

  1. The background part of section II lacks insights into the process of power conversion into fuel.

We added a bridging text citing a review paper before we switch to the “special” power-to-fuel concept with which we deal in the present work. By this way, we first introduce the power-to-fuel concept in general, and switch to the co-electrolysis based process afterwards.

  1. The method proposed in 3.6 for technical and economic evaluation has been relatively mature and has no novelty.

In this section (3.6) we assessed the technological maturity of the proposed configuration based on the well-established approach of technology readiness level. Based on the remark of the reviewer, we understand that that our approach was not clear. Therefore, we rephrased the text in several parts to explain what we did. We simply assessed each process step in our process based on the literature data for justification and at the end assessed the overall process based on the lowest TRL in the process chain.  

  1. Section 4 gives an overview of the current situation of fluid power system. It is suggested to increase the length and add some schematic diagrams to illustrate the demonstration process for readers' understanding.

Thank you very much for this remark. Based on your comment, we decided that this section matches better to the background section in its new version. Therefore, we shifted it to the front part.

  1. Section 4 does not fully reflect the "verification of the technical maturity of each component in the developed fuel synthesis", which should be supplemented.

As mentioned above, section 4 is shifted to the background part reflecting the state-of-art. The technical maturity is assessed in section 3.6.

  1. Section 6 places more emphasis on the discussion part. If possible, please introduce the simulation in detail and compare the system response to support this proposition

The simulation details (in 7 pages) are given in new Section 4. In the new section 5 we present and discuss the results from the process analysis focusing on fuel properties, process balance, evaluation of the heat recovery potential for CO2 separation and finally compare the results with those in the literature for low-temperature electrolysis to highlight the synergy of the proposed approach. Therefore, we would like to keep both parts separated, given the already high length of the paper.

  1. In Section 8, only one point is put forward. For the theme of future research, it is suggested to make more prospects for the fields to be further studied.

The conclusions were extended by points addressing the role and the future prospects of the CO2 separation technology. Furthermore, some aspects from the comparison part were shifted to the conclusions.

  1. Further elaboration of the images in Appendix A is recommended.

We explained the (only) image in Appendix A further. Moreover we introduced a cross link to this figure in Chapter 3.2.

Round 2

Reviewer 2 Report

In this paper, a "A techno-economic assessment of Fischer-Tropsch fuels based on syngas from co-electrolysis" is proposed. This study focuses on the coupling of Fischer-Tropsch synthesis for the production of synthetic diesel and kerosene to a high-temperature electrolysis unit. As a part of worldwide efforts to substantially reduce CO2 emissions, power-to-fuel technologies offer a promising path to make the transport sector CO2-free, complementing the electrification of vehicles. The author's research needs to be improved, and the following suggestions are put forward:

  1. In the introductory part of the article, there is an incompleteness in the descriptive logic of the statements, for example. “on the one hand…”. In addition, please consider carefully whether the section descriptions about the article make sense?
  2. In the section II, please use careful discretion as to whether citations to the same reference can be integrated accordingly.
  3. In section 3, it is proposed to reduce the description of some basic concepts, e.g. electrolysis. In the formulae, the meaning of the symbols is not fully explained.
  4. Why is the image a double copy?
  5. Further deliberation on the logic of the text and the repetition of phrases is recommended.

Author Response

Dear Reviewer,

Thank you very much for your critical remarks. In this revision round, we modified the manuscript further based on your suggestions.

First remark of the Reviewer:

In the introductory part of the article, there is an incompleteness in the descriptive logic of the statements, for example. “on the one hand…”.

Response: The introductory part of the article was revised in wording taking the concrete suggestion into account, but also with further corrections to remove the incompleteness issues mentioned.

Second remark of the Reviewer:

In addition, please consider carefully whether the section descriptions about the article make sense?

Response: The section descriptions were revised. The old structure was removed and the descriptions were adapted to the new structure of the article.

Third remark of the Reviewer:

In the section II, please use careful discretion as to whether citations to the same reference can be integrated accordingly.

Response: Section 2 was modified accordingly and citations to the same reference were integrated, especially at the front part of the section.

Fourth remark of the Reviewer:

    In section 3, it is proposed to reduce the description of some basic concepts, e.g. electrolysis. In the formulae, the meaning of the symbols is not fully explained.

Response: We revised the text so that we skipped the general description electrolysis in the beginning of Section 3.1. We directly step in with three electrolysis options and position the SOEC technology afterwards. We would like to keep the remaining descriptions for the sake of completeness.

The missing symbols from the formulae, which were not directly referred to in the original version, were integrated into the text to increase clearness.

Fifth remark of the Reviewer:

Why is the image a double copy?

Response: Unfortunately, since the Word version with track changes was used to generate the pdf version, the pdf file contains two images for every image we replaced in the revision. This replacement was required, since the quality of the images were reduced automatically during saving the file. We believe that this problem will not occur again in the second revision stage, since we have not made any further changes to images.

Sixth remark of the Reviewer:   

Further deliberation on the logic of the text and the repetition of phrases is recommended.

Response: We went through the text once more and modified several parts to remove wrong use of the phrase “on the other hand” and to increase the text flow to achieve better readability. 

Round 3

Reviewer 2 Report

In this paper, a "A techno-economic assessment of Fischer-Tropsch fuels based on syngas from co-electrolysis" is proposed. This study focuses on the coupling of Fischer-Tropsch synthesis for the production of synthetic diesel and kerosene to a high-temperature electrolysis unit. As a part of worldwide efforts to substantially reduce CO2 emissions, power-to-fuel technologies offer a promising path to make the transport sector CO2-free, complementing the electrification of vehicles. The author's research needs to be improved, and the following suggestions are put forward:

  1. Explain the names of the other components in Figure 2 and Figure 3.
  2. In the sensitivity analysis, it is recommended to add some sensitivity analysis charts so that the sensitivity factors can be visualized.
  3. In Appendix B, Table 11 suggests further adjustments to avoid confusion.
  4. Why is the image a double copy?
  5. Some of the references are too old, can they be considered for replacement?

Author Response

Dear Reviewer,

Thank you very much for your further remarks, as well as time and effort. We modified the manuscript further based on your suggestions. Please find our responses to your suggestions below.

  1. Explain the names of the other components in Figure 2 and Figure 3.

The names of the remaining components in Figures 2 and 3 are explained.

  1. In the sensitivity analysis, it is recommended to add some sensitivity analysis charts so that the sensitivity factors can be visualized.

The visualization of the sensitivity factors was realized in Figure 21. The best case and the worst case factors are now highlighted with the corresponding color (green for best case and red for worst case) explicitly on the left part of the figure, so that the reader can directly have a feeling with which assumption the best and the worst case price is achieved. The original version was not self-explaining. Since the detailed influence of the single values are already discussed and presented in the sections 6.4.1 to 6.4.5 and Figures 16 to 20, and the same maximum and minimum values are used here, we prefer not to add additional figures for the sensitivity analysis.

  1. In Appendix B, Table 11 suggests further adjustments to avoid confusion.

We explained what a negative sum means for the process steam balances and made a cross reference to Table 10 where the resources used in this table are explained in detail.

  1. Why is the image a double copy?

As in the previous version, since the journal requires a version of the manuscript in the track changes mode to trace the changes and this version is automatically used to generate the pdf version for review purposes, the images appear twice. We unfortunately do not have an influence on this.

  1. Some of the references are too old, can they be considered for replacement?

The oldest reference is from 1987. We checked if we could replace this source, but unfortunately found out that this is very valuable source, which analyzed fuel properties in detail. Since this was the main source for our related statement, we decided to keep the source in order to credit the original source appropriately.

The second oldest reference was from 1995 and it was referring to the TRL method and that it was first found by NASA. We could replace this source with a newer source from NASA.

The third oldest reference is from 1996. The author of the publication is from Aspen Tech and the source has received a very well resonance, therefore, it was worth keeping this source, which was also critical for us.
